# Boosting spontaneous orientation polarization of polar molecules based on fluoroalkyl and phthalimide units

Masaki Tanaka ◉ ✉

Polar organic molecules form spontaneous polarization in vacuum-deposited films by permanent dipole orientations in the films, originating from the molecule's potential ability to align itself on the film surface during deposition. This study focuses on developing polar molecules that exhibit spontaneous orientation polarization (SOP) and possess a high surface potential. In the proposed molecular design, a hexafluoropropane (6F) unit facilitates spontaneous molecular orientation to align the permanent dipoles, and a phthalimide unit induces strong molecular polarization. Furthermore, the introduction of phthalimides into the molecular backbone raises the glass transition temperature of the molecules, leading to the suppression of molecular mobility on the film surface during film deposition and an improvement in the dipole orientation. The resulting surface potential slope is approximately 280 mV nm⁻¹ without substrate temperature control. Furthermore, this work proposes a method using position isomers as a design strategy to tune the SOP polarity. The substitution position of the strong polar units influences the direction of the total molecular dipoles and affects the SOP polarity of the 6F-based molecules. The proposed molecular designs in this study provide wide tunability of the SOP intensity and polarity, which contributes to highly efficient organic optoelectronic and energy-harvesting devices.

Precise control of molecular orientations in organic thin films is a critical requirement to achieve ultimately high-performance organic devices, such as organic light-emitting diodes (OLEDs), organic photovoltaics, and organic field-effect transistors, because charge mobilities and light-emission performances strongly depend on the molecular orientations[1-3]. Spontaneous orientation polarization (SOP) is derived from ordered metastable orientation states with aligned permanent dipole moments (PDMs) of polar molecules in solids[4-7]. Then, the deposited film of tris(8-hydroxyquinolinato) aluminium (Alq₃) exhibited a positive surface potential (giant surface potential; GSP) of +28 V at a film thickness of 560 nm, indicating that the growth rate of the GSP to film thickness (GSP slope) was +50 mV nm⁻¹ (Fig. 1a)[4]. Previous studies have revealed that SOP formation is not a unique event for Alq₃ deposition but a universal event for most amorphous-type polar molecules in their vacuum deposition processes[8]. On the other hand, there are negative surface charges on the bottom interface of positive GSP films[9], that is, the emission layer (EML)/electron-transport layer (ETL) interface of OLEDs with SOP molecules such as Alq₃ as an ETL. The negative surface charges induce hole accumulation at the interface and cause exciton quenching via exciton-polaron annihilations leading to inferior device performance such as quantum yields and operational stabilities[10,11]. Furthermore, SOP promotes charge separation of charge-transfer exciplex excitons between organic donor and acceptor molecules, which would be beneficial for improving the performance of organic photovoltaics and organic photodetectors[12]. Additionally, an electret material for vibration power generators is another application of SOP films because a spontaneously formed dipolar film can be used as a self-assembled electret

Department of Biotechnology and Life Science, Tokyo University of Agriculture and Technology, Koganei, Tokyo, Japan. ✉e-mail: m-tanaka@me.tuat.ac.jp

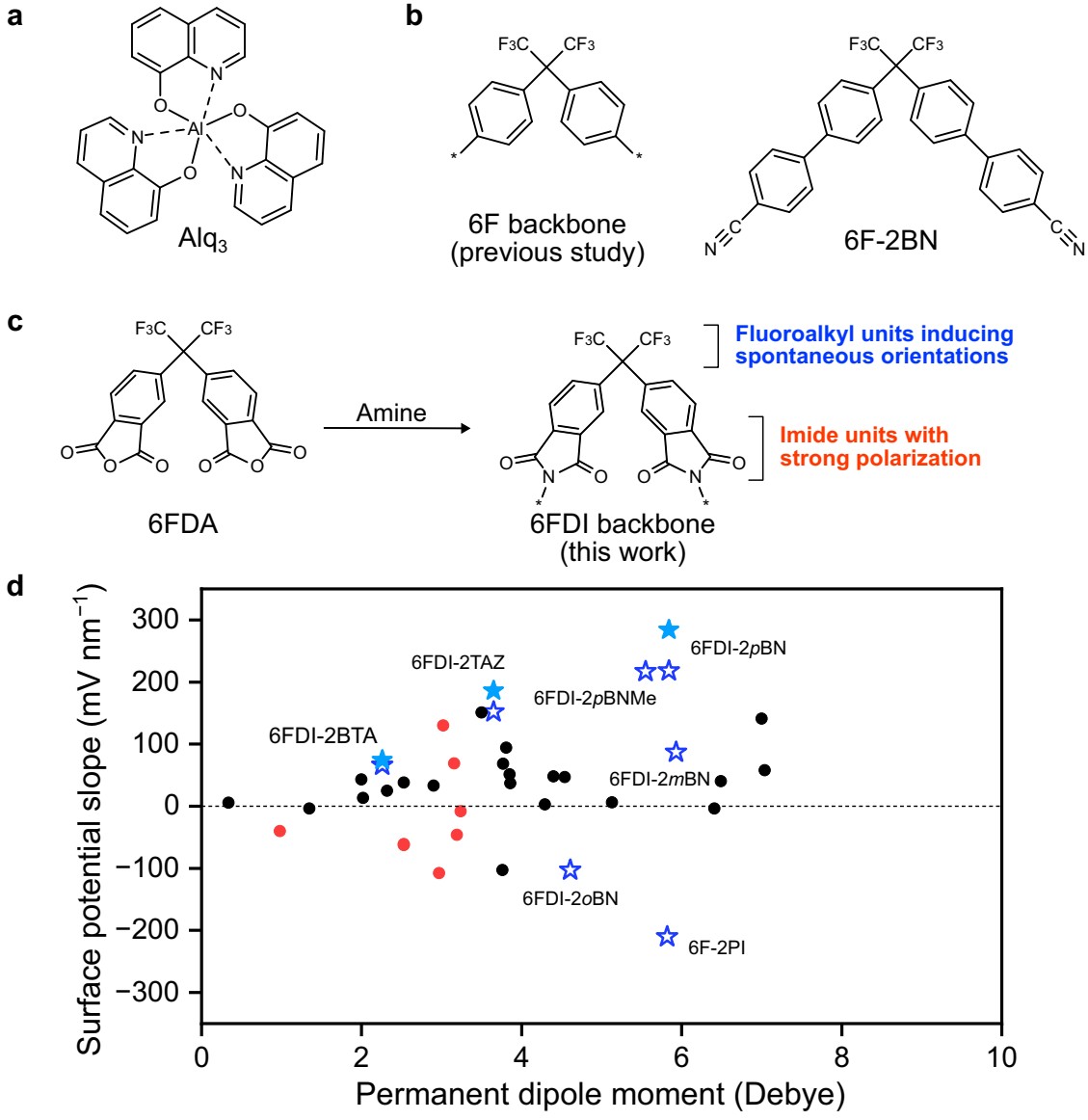

**Fig. 1 | Molecules exhibiting spontaneous orientation polarization. a** Alq₃. **b** 6F backbone and 6F-2BN. **c** 6FDI backbone. **d** Relationship between permanent dipole moment and the reported surface potential slope values of vacuum-deposited films[5,19,24]. Red symbols represent the slope values of previously reported 6F-based molecules[22]. Star symbols represent the slopes of the molecules developed in this study. The filled star symbols represent the slope values of 6FDI-2BTA, 6FDI-2TAZ, and 6FDI-2pBN films deposited at high deposition rates.

(SAE)[13,14]. Thus, SOP can provide a unique perspective on the performance of organic devices, which differs from the widely investigated molecular packings or orientations that lead to high charge mobilities and efficient light out-coupling. Therefore, to improve and precisely control the performance of these optoelectronic and energy-harvesting devices, the optimization of the SOP intensity and polarity is critical. However, a design strategy for SOP molecules has not been fully established, because the dipole orientation mechanism is not entirely clear.

The film polarization ($P$) formed by the spontaneous dipole orientations can be expressed as $P = p\langle\cos\theta\rangle n$, where $p$, $\langle\cos\theta\rangle$, and $n$ are the PDM of polar molecules, the mean orientation degree of PDMs, and the molecular density of films, respectively. Thus, improvement of $\langle\cos\theta\rangle$, that is, the molecular orientation with the same direction toward the film growth direction, is essential for the enhancement of SOP because film polarization is cancelled when polar molecules form the inverted direction of the PDM orientation on the film surface by dipole-dipole interactions[8,15]. The $\langle\cos\theta\rangle$ values of typical SOP

molecules, such as Alq₃ and 1,3,5-tris(1-phenyl-1H-benzo[d]imidazol-2-yl)benzene (TPBi), were less than 0.1, indicating that molecular PDMs cannot efficiently contribute to dipolar film formation. Recent studies have proposed that the intrinsic SOP formation mechanism involves anisotropic van der Waals interactions between deposited molecules and film surfaces[16,17]. Furthermore, process factors, such as a deposition rate, a deposition step, and a substrate temperature, also affect $\langle\cos\theta\rangle$ values and signs of SOP[18–21]. The author's group recently developed fluoroalkyl-based polar molecules (Fig. 1b) that exhibit high $\langle\cos\theta\rangle$ values of over 0.3[22]. However, the fluoroalkyl-based molecular backbone for SOP formation and the GSP slope value are still limited to realizing high-performance organic optoelectronic and energy-harvesting devices. In this study, a 4,4′-(hexafluoroisopropylidene) diphthalic imide (6FDI) backbone (Fig. 1c) was used to develop polar molecules with a strong molecular PDM. The strong molecular polarization of fluoroalkyl-based molecules contributed to the formation of polarized films with a high GSP slope of over 200 mV nm⁻¹ on a substrate without substrate temperature control (Fig. 1d, Supplementary

Table 1). The proposed molecular design helps to improve the SOP and achieve high-performance organic electronic and energy-harvesting devices.

## Results and Discussion

Fluoroalkyl units in molecules induce spontaneous orientations in amorphous films owing to their small surface free energy, which the fluoroalkyl units preferentially face on the film surface side (vacuum-side)[22]. The small polarizability results in weakend van der Waals interactions between the units and the film surface during the deposition process, then, the fluoroalkyl units such as the $CF_3$-units escape from the interface between the deposited molecule and the film surface to face the vacuum side. Although the previous study used 2,2-diphenylhexafluoropropane (6F) as the basic backbone for the proposed $CF_3$-based molecular design, this study used the 6FDI structure to design SOP molecules with a strong molecular PDM. The 6FDI molecules were synthesized via a one-step chemical reaction of 4,4′-(hexafluoroisopropylidene)diphthalic anhydride (6FDA) and amine molecules with different aromatic groups. The end groups of the 6FDI molecules can be modified by applying a variety of amine molecules to the chemical reaction. In this study, acceptor-type end groups such as benzothiazole (BTA), triazole (TAZ), and $p$-benzonitrile ($p$BN) were introduced, and 6FDI-2BTA, 6FDI-2TAZ, and 6FDI-2$p$BN were synthesized (Fig. 2a). The calculated PDM vector of the 6FDA molecule using the density functional theory (DFT) method indicates that the $CF_3$-side is positively polarized in the molecule owing to the strong polarization of phthalic anhydride. The dipole directions of the 6FDI molecules are almost the same as those of 6FDA, that is, the positively polarized $CF_3$-side in the molecules. The author notes that there are several stable conformers of these molecules (Fig. 2b), and the effect of the conformers is discussed in a later section.

Figure 3a shows the dependence of the surface potentials of the vacuum-deposited films on the film thickness. The molecules were then deposited on indium tin oxide (ITO)-coated glass substrates. The substrate temperature ($T_s$) was room temperature, and the deposition rate of the molecules was approximately 0.1 nm s$^{-1}$. Although a 6FDA film showed almost no change in surface potentials (GSP slope: +0.1 mV nm$^{-1}$), the 6FDI films exhibited positive GSPs. The GSP slopes of 6FDI-2BTA, 6FDI-2TAZ, and 6FDI-2$p$BN were +66 mV nm$^{-1}$, +152 mV nm$^{-1}$, +218 mV nm$^{-1}$, respectively (Table 1). The positive GSPs indicate that the positively polarized fluoroalkyl-side of the molecules averagely faced the vacuum-side and formed vacuum-deposited films

with ordered molecular PDMs[22]. Interestingly, 6FDI-2$p$BN exhibited a significantly high GSP slope compared to 6F-2BN (GSP slope: +69 mV nm$^{-1}$) which has a molecular structure similar to that of 6FDI-2$p$BN.

Furthermore, the GSPs were improved by vacuum deposition at a higher deposition rate. Figure 3b shows the thickness dependence of the GSP slopes of the developed molecules deposited at a deposition rate of 0.26–0.29 nm s$^{-1}$. The developed 6FDI molecules also exhibited a higher GSP slope at a high deposition rate, indicating an improved degree of orientation of the PDMs (Table 1). In particular, the deposited 6FDI-2$p$BN film exhibited a high surface potential of +284 mV nm$^{-1}$. Thus, the 6FDI-2$p$BN film deposited at a $T_s$ of RT achieved a GSP slope comparable to that of a film of bis-4-(N-carbazolyl)phenylphosphine oxide deposited at a $T_s$ of −70 °C[19]. Note that the surface charge density ($\sigma$) of the dipolar films (Table 1) were calculated using the following equation, $\sigma = $ (GSP slope) $\times \varepsilon_r \times \varepsilon_0$, where $\varepsilon_r$ and $\varepsilon_0$ are the relative permittivity, the dielectric constant of vacuum. The $\varepsilon_r$ value was assumed to be 3.0 in all organic films[23].

Generally, organic molecules possess several stable conformers with different PDM magnitudes and directions. Because the average PDM in a molecule directly affects the magnitude and polarity of the SOP, conformation control is essential for improving the SOP[19,24–26]. The possible conformations were searched using the force-field theory, and the obtained molecular structures were optimized using the DFT method. Although 6FDA and 6FDI-2$p$BN had only three stable conformations, the number of conformers of 6FDI-2BTA and 6FDI-2TAZ was 62 and 10, respectively, and the distribution of the PDM of the conformers and their populations are shown in Fig. 4. Thus, the structural asymmetry of the introduced end-groups was correlated with the number of conformers, relating to the average PDM magnitude and the direction. The average PDM magnitude, $\langle p \rangle$, was calculated as the sum of the production of the PDM magnitude and the population of each conformer. The $\langle p \rangle$ values of 6FDA, 6FDI-2BTA, 6FDI-2TAZ, and 6FDI-2$p$BN were 3.43, 2.26, 3.65, and 5.84 Debye, respectively. The $\langle p \rangle$ of 6FDI-2$p$BN was larger than that of 6F-2BN because of the introduced PI groups, which was one of the reasons for the higher GSP slope of 6FDI-2$p$BN.

Additionally, an average orientation degree of PDMs, $\langle \cos \theta \rangle$, was calculated based on the following equation, $P = \langle p \rangle \langle \cos \theta \rangle n$. The calculated $\langle \cos \theta \rangle$s of the 6FDI molecules were approximately 0.2, which were higher than that of the previously reported 6F-2BN (0.10–0.17), despite the presence of the several 6FDI conformers with the

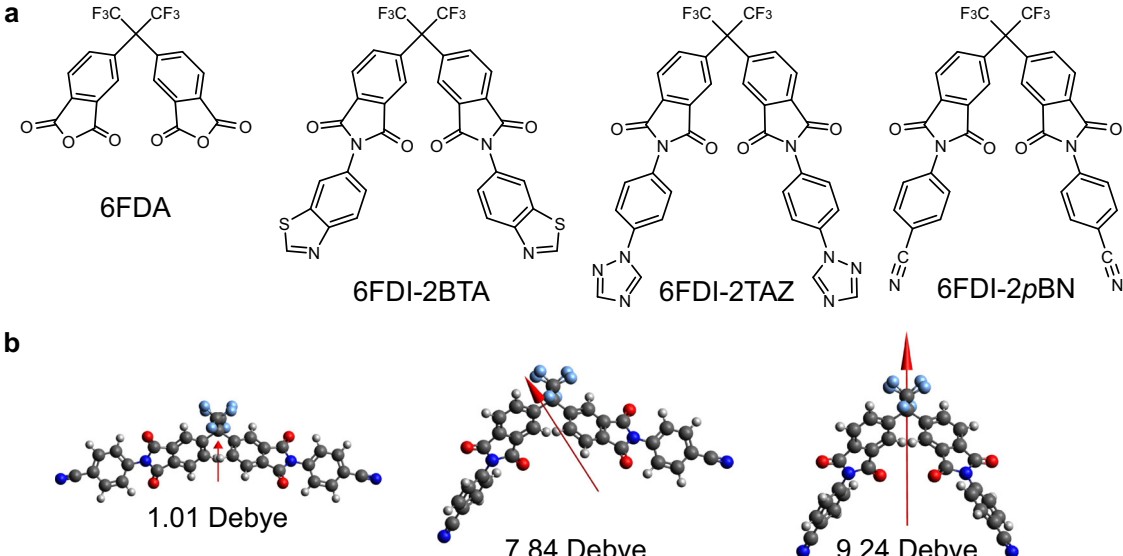

**Fig. 2 | 6FDI molecules. a** Molecular structures of 6FDA, 6FDI-2BTA, 6FDI-2TAZ, and 6FDI-2$p$BN. **b** Stable conformational structures of 6FDI-2$p$BN.

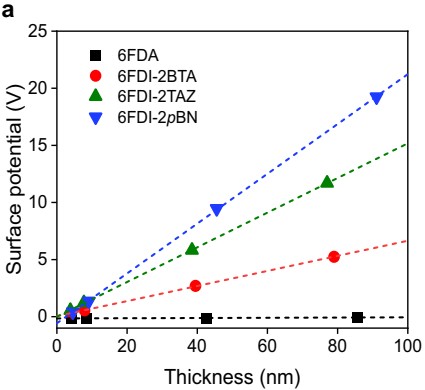
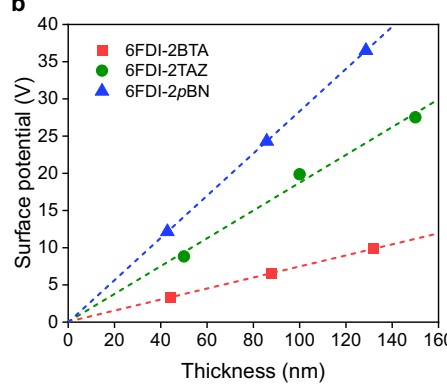

**Fig. 3 | Surface potentials of vacuum-deposited films. a** Thickness dependence of the surface potentials of the developed 6FDI molecules with a deposition rate of approximately 0.1 nm s$^{-1}$. **b** Thickness dependence of the surface potentials of the deposited films at higher deposition rates.

dispersed dipole directions. Therefore, the notably high GSP slopes of the 6FDI molecules originate not only from the larger $\langle p \rangle$, but also from the improved PDM orientations. The $\langle \cos\theta \rangle$ of 6FDI-2BTA was relatively small compared to the other 6FDI molecules because of the presence of conformers with inverted PDM directions owing to the structural asymmetry of the BTA group. The positive and the negative dipoles of the conformers (Fig. 4a, b) resulted in the cancellation of molecular dipoles in the deposited films, leading to a reduction in $\langle \cos\theta \rangle$. Nevertheless, methyl-substituted 6FDI-2*p*BN (6FDI-2*p*BNMe) exhibited a GSP slope and a $\langle \cos\theta \rangle$ value comparable to that of 6FDI-2*p*BN despite the low structural symmetry of the end-group, leading to an increase in the number of conformers (Supplementary Fig. 1). This is because the methyl-substitution has a minimal impact on the PDM direction, and there are only a few conformers with a negative dipole reducing the overall $\langle \cos\theta \rangle$. Therefore, the management of the PDM direction (positive or negative) rather than the number of conformers is more important for improving the $\langle \cos\theta \rangle$ of 6FDI molecules.

The glass transition temperature ($T_g$) of deposited molecules is one of the factors related to the molecular orientations[27–29]. The ratio of $T_s$ to $T_g$ ($T_s/T_g$) correlates with the molecular surface mobility on the deposited surface. Excess molecular mobility interrupts the formation of parallel dipole orientations, leading to the formation of anti-parallel or random orientations with reduced SOP. A low $T_s/T_g$, limiting the molecular mobility, is beneficial for accelerating the formation of a parallel dipole orientation to improve the SOP. The suppressed PDM orientation of 6F-2BN originates from its relatively low $T_g$ (74 °C). The $T_g$ values measured using differential scanning calorimetry (DSC) of

the 6FDI molecules were over 130 °C, which was significantly higher than that of 6F-2BN (Table 1 and Supplementary Fig. 2). Therefore, the limited mobility of the 6FDI molecules contributes to an improvement in the orientation degree. Additionally, $T_s$ during depositions was also controlled using a heater connected to a substrate holder to validate the effect of $T_s/T_g$ on the GSP slopes. The $T_s$ dependence of the GSP slopes of the deposited 6F-2BN and 6FDI-2*p*BN films is shown in Supplementary Fig. 3. The GSP of the 6FDI-2*p*BN film was maintained at a higher $T_s$, however, that of the 6F-2BN film dramatically decreased with an increase in $T_s$, and the GSP became almost zero at a $T_s$ of 45 °C during the deposition. Thus, the lower $T_g$ of 6F-2BN resulted in the critical $T_s$ dependence of the GSP slopes in these $T_s$ ranges over room temperature. Therefore, the higher $T_g$ is one of the reasons for the significant improvement in the orientation degree and the GSP slope of 6FDI-2*p*BN. Furthermore, it is expected that a decrease in $T_s$ below room temperature enhances the GSP slopes of these molecules, owing to a low $T_s/T_g$ with further limited molecular mobility, as reported previously[19]. The author estimated that one of the reasons for the zero polarization of a vacuum-deposited 6FDA film was the low $T_g$ (61 °C) corresponding to a high $T_s/T_g$ of 0.89 at room temperature, resulting in the randomization of molecular orientations. Hence, cooling $T_s$ is expected to build a positive GSP of a 6FDA film.

Another factor limiting the GSP slope and the $\langle \cos\theta \rangle$ value, except for molecular PDM and low $T_g$, is intermolecular dipole-dipole interactions between polar molecules on the film surface during film formation. The intensity of dipole interactions depends on the PDM of polar molecules, indicating that polar molecules with strong PDM are considerably influenced by dipole interactions, leading to the formation of anti-parallel dipole pairs to reduce $\langle \cos\theta \rangle$ and film polarization. Thus, the GSP slope of conventional polar molecules is limited, despite the strong molecular PDM (black symbols in Fig. 1d). In contrast, 6FDI-2*p*BN with $\langle p \rangle$ of 5.84 Debye maintained the moderate $\langle \cos\theta \rangle$ and the high GSP slopes. This was attributed to the vertical molecular orientation induced by the 6F-units. Thus, the 6F-backbone is useful for designing SOP molecules with strong PDM to achieve extremely high GSP slopes.

The SOP polarity design of previously reported 6F molecules was successfully performed using a wide variety of functional groups with different polarity directions. In this study, the design of position isomers is introduced to control SOP polarity, that is, positive and negative GSPs. As shown above, the BN group induces strong polarization, in which the nitrogen atom is negatively polarized. Thus, the positional isomers of the BN group are useful for controlling the direction of the molecular PDM. The positional isomers of 6FDI-2*p*BN, that is, 6FDI-2*o*BN and 6FDI-2*m*BN (Fig. 5a), also possess several conformers with different molecular PDMs. The calculated results of the

## Table 1 | Summary of the properties of polar films of the developed molecules

| | $T_g$[a] (°C) | Deposition rate (nm s$^{-1}$) | $\langle p \rangle$[b] (Debye) | GSP slope (mV nm$^{-1}$) | $\sigma$[c] (mC m$^{-2}$) | $\langle \cos\theta \rangle$[d] |
|---|---|---|---|---|---|---|
| 6F-2BN[e] | 74 | 0.1 | 3.16 | +69 | +1.8 | +0.10 |
| | | 0.3 | | +117 | +3.1 | +0.17 |
| 6FDI-2BTA | 139 | 0.08 | 2.26 | +66.0 | +1.75 | +0.13 |
| | | 0.26 | | +74.2 | +1.97 | +0.15 |
| 6FDI-2TAZ | 151 | 0.08 | 3.65 | +152 | +4.03 | +0.19 |
| | | 0.29 | | +186 | +4.93 | +0.23 |
| 6FDI-2*p*BN | 132 | 0.09 | 5.84 | +218 | +5.78 | +0.17 |
| | | 0.26 | | +284 | +7.52 | +0.22 |

[a]Glass transition temperature.
[b]Average permanent dipole moment magnitude.
[c]Surface charge density.
[d]Average orientation degree
[e]Tanaka et al.[22].

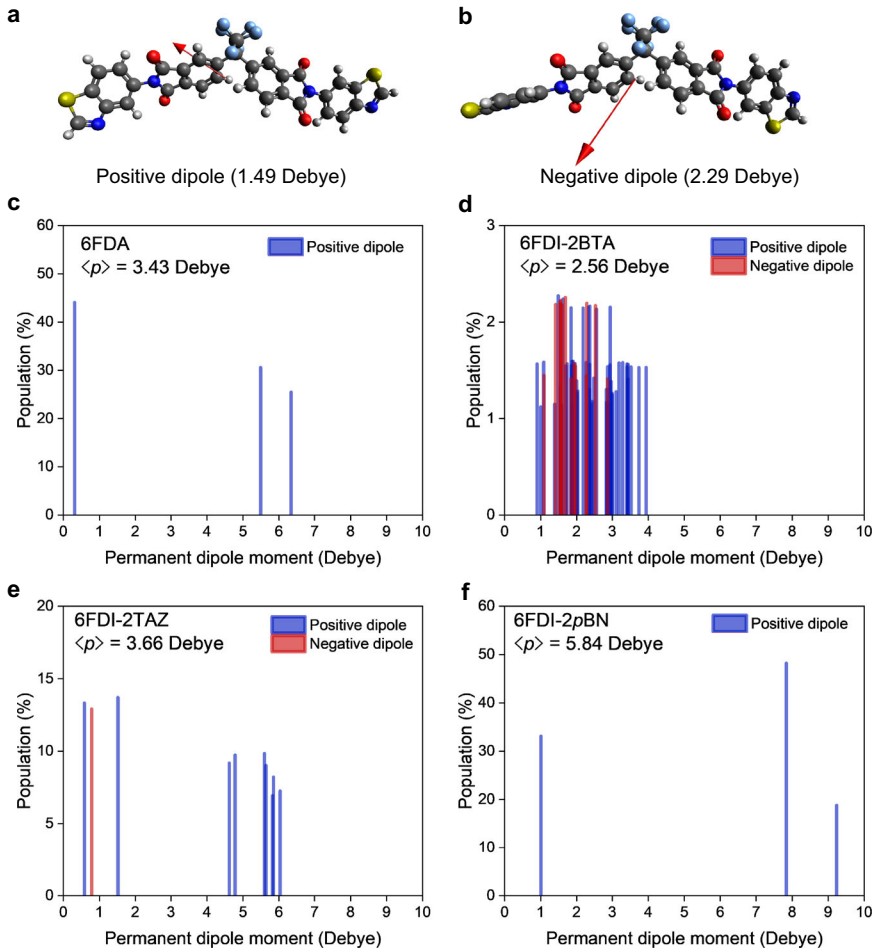

**Fig. 4 | Permanent dipole moment of molecular conformers.** Molecular conformations of 6FDI-2BTA with **a** positive and **b** negative dipoles. Relationships between the permanent dipole moment and the population of the calculated conformers of **c** 6FDA, **d** 6FDI-2BTA, **e** 6FDI-2TAZ, and **f** 6FDI-2pBN.

PDM of the conformers indicated that the majority of the PDM direction of 6FDI-2mBN (Supplementary Fig. 4a) was the same as that of 6FDI-2pBN, indicating that the 6F moiety of the molecules was positively polarized, leading to a positive GSP. In contrast, most conformers of the o-isomer, 6FDI-2oBN (Supplementary Fig. 4b), possess an inverted PDM direction with 6FDI-2pBN, indicating the negatively polarized 6F part in the molecules, leading to a negative GSP. The measured GSPs of the vacuum-deposited films of the positional isomers are shown in Fig. 5b. As predicted from the calculation results, the GSP polarities of 6FDI-2mBN and 6FDI-2oBN were positive and negative, respectively (Fig. 5c). These results clearly indicate that GSP polarity does not originate from native molecular properties, such as molecular mass, elemental ratio, and functional groups. Although the number of negative SOP molecules is still limited except for the $CF_3$-based molecules, it is essentially required to consider orientation manners and molecular PDM directions to understand the clear origin of the SOP polarity.

The author estimated that one of the reasons for the smaller absolute values of the GSP slopes of 6FDI-2oBN and 6FDI-2mBN is the wide distribution of the PDM magnitude and the direction, resulting in the smaller effective PDM magnitude of the 6FDI molecules. To reduce the effects of conformers on the PDM orientations, a molecule with the 6F-backbone and the PI groups, 6F-2PI, was also prepared (Fig. 5a). According to conformation analysis, 6F-2PI has one molecular conformation with a PDM magnitude of 5.82 Debye (Supplementary Fig. 4c). The GSP polarity of 6F-2PI (Fig. 5b) was negative, and the slope was −210 mV nm⁻¹, corresponding to an orientation degree of −0.17,

which was significantly higher than that of 6FDI-2oBN (Supplementary Table 2). This is attributed to the preferable PDM direction along the center of the 6F-backbone, leading to an efficient dipole alignment toward the substrate in the normal direction.

SOP films can be applied as SAEs in vibration-based energy harvesters[13,14]. Although the stability of GSP is necessary for electret applications, conventional SOP molecules, such as $Alq_3$, absorb visible light to cancel GSPs. The developed polar molecules have no clear absorption in the visible light region (Supplementary Fig. 5); thus, the GSPs can potentially be maintained under visible-light irradiation. The thermal stability of the GSPs was discussed in a previous paper[22], and the $T_g$ of polar molecules is one of the limitations of the electret's heat-resisting temperature because the GSP vanishes by orientation randomization in the deposited films over $T_g$. Compared to 6F-2BN, the $T_g$ values of the 6FDI molecules were relatively high (>130 °C), indicating that improved thermal stability can be expected. To verify the operation of the electret film of 6FDI-2pBN, the current generation was demonstrated using a vibration probe of the KP system[13]. Supplementary Fig. 6 shows the current profile generated via probe vibration above the deposited 6FDI-2pBN film (surface potential of ~20 V), indicating that the deposited film of the developed molecules is applicable for energy harvesters and vibration sensors as presented in previous studies[13,14].

The energy levels of the highest occupied molecular orbital (HOMO) were determined to be 7.1–7.9 eV (Supplementary Table 3) using a photoemission yield spectrometer (Supplementary Fig. 7). The lowest unoccupied molecular orbital (LUMO) levels were estimated

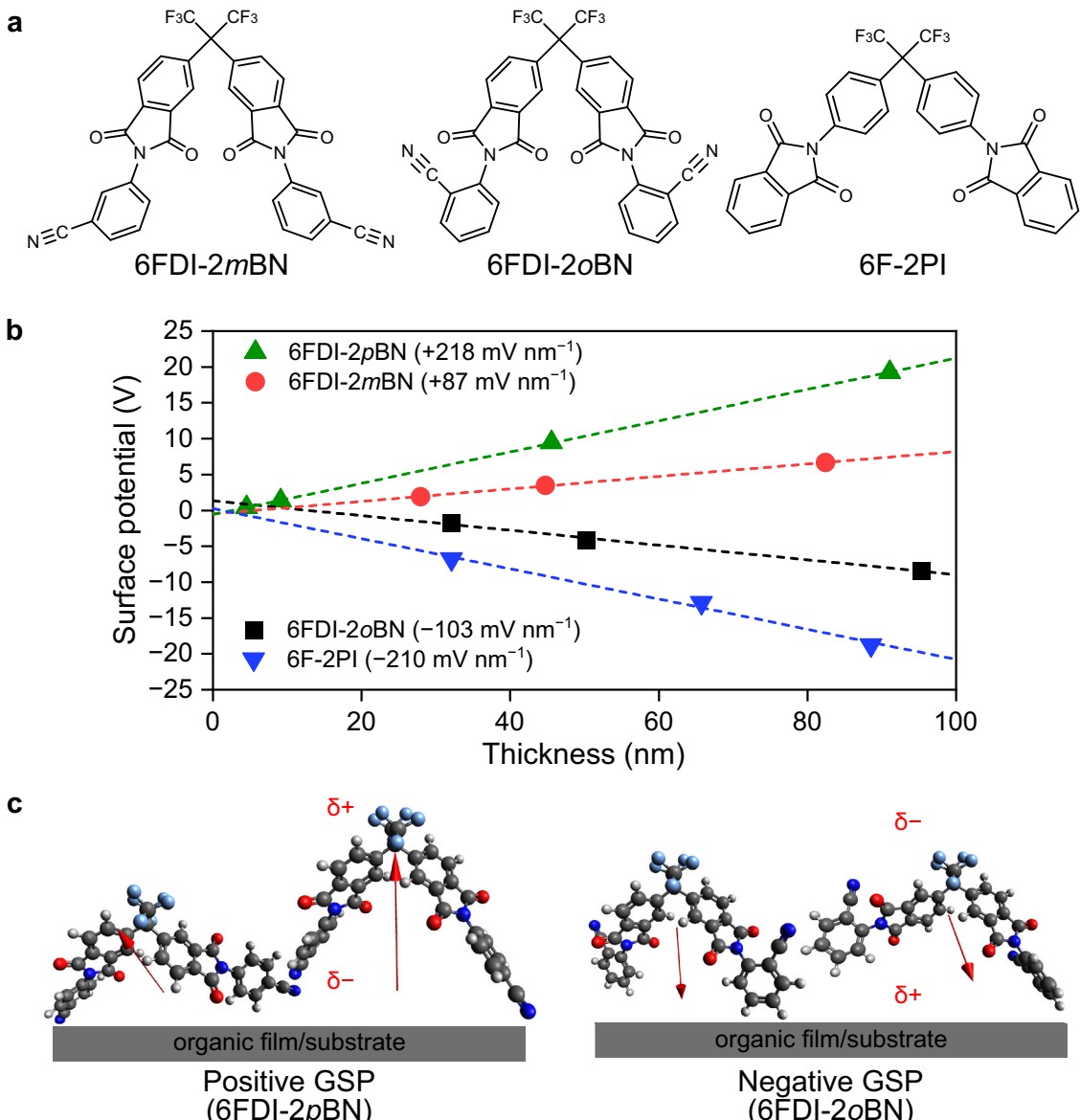

**Fig. 5 | Polarity tuning of orientation polarization by molecular design.** **a** Molecular structures of 6FDI-2oBN, 6FDI-2mBN, and 6F-2PI. **b** Thickness dependence of surface potentials of developed molecules. **c** Schematic of the formation of spontaneous orientation polarization with different polarities of 6FDI molecules on an organic film or a substrate.

using the HOMO levels and the optical gap values (Supplementary Table 3). The developed molecules possess deep HOMO and LUMO levels compared to typical electron transport molecules, such as Alq₃, because of the acceptor-rich 6F-based molecular structures. The vacuum-deposited films exhibited low-intensity photoluminescence (PL) (Supplementary Fig. 8a–g). The shapes of the PL spectra were relatively broad, and the PL peak wavelength was approximately 500 nm. These are mainly attributed to charge-transfer (CT)-type emissions because the HOMO and LUMO distributions are spatially separated in the molecules. (Supplementary Fig. 8h). The carrier transport properties of the developed positive and negative SOP molecules, such as 6FDI-2pBN and 6FDI-2oBN, were characterized using hole-only and electron-only devices (HODs and EODs; device structures are shown in Supplementary Fig. 9a, b). The zero-field hole and electron mobilities ($\mu_{0h}$ and $\mu_{0e}$) were estimated using the Child's law, that is, $J = \frac{9}{8}\mu_0\varepsilon_r\varepsilon_0\frac{V^2}{d^3}$, where $\mu_0$ and $d$ are the zero-field carrier mobility and thickness of organic films, respectively[30]. The calculated $\mu_{0e}$ and $\mu_{0h}$ were $1.1–5.7\times10^{-9}$ and $6.8–7.6\times10^{-10}$ cm² V⁻¹ s⁻¹, respectively (Supplementary Table 4). The author estimates that the low carrier mobilities are attributed to the broad distribution of the density of states (DOS) owing to dipolar disorder and/or molecular conformations[31]. Previous studies have revealed that large PDMs in deposited films induce a broad DOS, lowering the carrier mobilities[16,32]. Furthermore, as discussed above, the 6FDI-backbone possesses several molecular conformations. Supplementary Figs. 10 and 11 show the computationally calculated HOMO and LUMO energy alignments of the different conformations. Their energy levels are slightly different, which leads to a broadening of the DOS and a decrease in the charge mobilities.

The author examined OLED performance using the developed dipolar films as ETLs. The OLED device structure (Supplementary Fig. 12a) based on a thermally activated delayed fluorescence emitter, 1,2,3,5-tetrakis(carbazol-9-yl)-4,6-dicyanobenzene (4CzIPN), was 1,4,5,8,9,11-hexaazatriphenylenehexacarbonitrile (HAT-CN; 10 nm) /

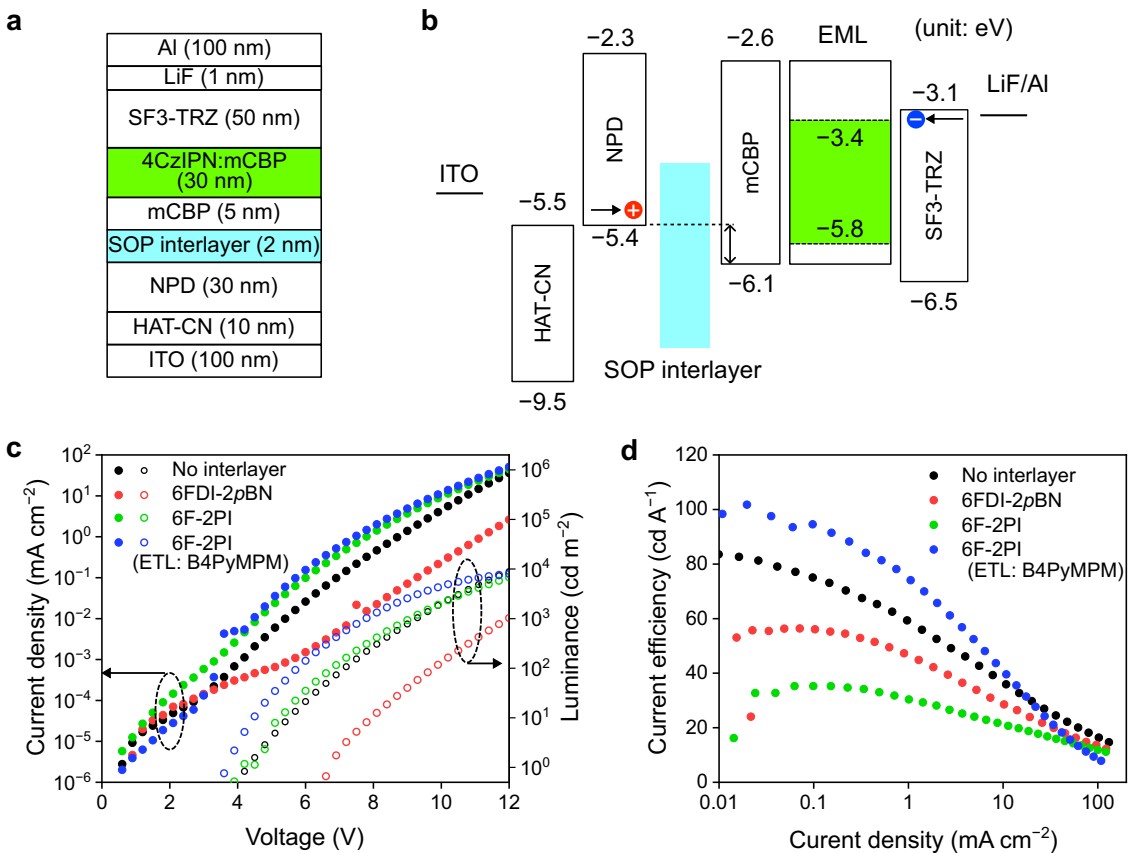

**Fig. 6 | OLEDs with SOP interlayers. a** OLED structure. **b** Energy level diagrams. **c** Current density–luminance–voltage characteristics. **d** Current efficiency profiles.

N,N′-di-1-naphthyl-N,N′-diphenylbenzidine (NPD; 30 nm)/3,3′-di(9H-carbazol-9-yl)-1,1′-biphenyl (mCBP; 5 nm)/15 wt% 4CzIPN:mCBP (30 nm)/ETL (50 nm)/LiF (1 nm)/Al (100 nm). 2-(9,9′-spirobi[fluoren]-3-yl)-4,6-diphenyl-1,3,5-triazine (SF3-TRZ), 6FDI-2oBN, and 6FDI-2pBN were used as ETLs with nonpolar, negative SOP, and positive SOP, respectively. Supplementary Fig. 12b–d shows the J–V-luminance (L), current efficiency (CE) profiles, and electroluminescence (EL) spectra of the OLEDs. The driving voltages of the OLEDs based on the 6FDI molecules were higher than that of the SF3-TRZ-based OLED. This was attributed to the low electron mobility of the 6FDI molecules (Supplementary Table 4) compared to SF3-TRZ[33]. Note that the deep LUMO levels of the 6FDI molecules allow electrons to be directly injected into doped 4CzIPN in the EML (Supplementary Fig. 12e). Because the EL spectra of the OLEDs mainly attributed to 4CzIPN emission and the delayed EL lifetimes (4.1 µs) approximately correspond to the delayed PL lifetime (3.5 µs) of 4CzIPN doped in mCBP, electrons are directly injected to 4CzIPN molecules and charge recombination occurs in the EML (Supplementary Fig. 13). Although the SF3-TRZ-based OLED exhibited a maximum CE of 83.6 cd A$^{-1}$ (corresponding to an external EL quantum efficiency of ~25%)[33,34], those of the OLEDs based on 6FDI-2oBN and 6FDI-2pBN were 30.2 and 1.9 cd A$^{-1}$. The reason for the low CE was attributed to exciton quenching at the EML/ETL interface. The low electron mobilities of the 6FDI-based ETL and the high electron injection barrier between the ETLs and the EMLs facilitate hole accumulation and highly concentrated charge recombination at the interface, resulting in severe exciton-exciton and exciton-charge annihilations[35–37]. Furthermore, the interfacial charge at the EML/ETL interface also induce charge accumulation. Supplementary Fig. 14a) shows the displacement current measurement (DCM) profiles (ramp rate: 2 kV s$^{-1}$) of the OLEDs, where the current was measured using a current amplifier under triangular voltage applications. These results

indicate that the ETLs with positive/negative SOPs induce the charge injection and accumulation at the lower voltage than the EL turn-on (Supplementary Fig. 14b, c). The negative/positive polarization charges at the EML/ETL interface induce hole/electron accumulations to induce exciton-charge annihilations lowering the EL quantum efficiency[11,18,38,39]. The author estimated that the reason for the more severe CE drop of the OLED based on the 6FDI-2pBN ETL is the positive charge accumulation at the interface. The accumulated holes induce highly dense cations of 4CzIPN and mCBP, which possess absorption overlap with the 4CzIPN emission[36], indicating that 4CzIPN excitons are efficiently quenched by the cations compared to anions.

A previous study investigated the impact of SOP layers on hole injection at the ITO/hole-transport layer interface and revealed that an introduced negative SOP layer improves the hole current of hole-only devices[22]. This could be attributed to the adjustment of the work function using a dipole layer that reduces the carrier injection barrier, which has been well studied using self-assembled monolayers (SAMs)[40]. Although SAM modifications can form a highly ordered and strong dipolar monolayer on metal electrodes, they are generally formed by a solution process method and are not applicable to organic/organic interfaces. On the other hand, the SOP layers formed by vacuum deposition are applicable to organic/organic and organic/metal interfaces. This study examined the impact of SOP interlayers on hole injection at organic/organic interfaces. In the OLED structure shown in Fig. 6a, the relatively large HOMO level gap (~0.7 eV) between the NPD and the mCBP layers lowers the hole injection at the interface (Fig. 6b). To investigate the impact of the SOP layer at organic/organic interfaces, 2-nm-thick SOP layers of the developed polar molecules were introduced as interlayers at the interface between the NPD and the mCBP layers (Supplementary Fig. 15a, b). The reason for the small thickness of the SOP interlayers is the low hole mobility and the deep

HOMO levels, reducing hole transport in OLEDs. Therefore, the author expects that the SOP interlayers act as interfacial dipoles to tune the interfacial energy differences[41]. Figure 6c shows the $J–V–L$ characteristics of OLEDs with SOP interlayers. The OLED with the 6FDI-2$p$BN (positive SOP) interlayer exhibited a high driving voltage, whereas the OLED with the 6F-2PI (negative SOP) interlayer exhibited a low driving voltage compared to the OLED without interlayers. Furthermore, the thickness dependence of the surface potential of the mCBP/SOP-interlayer/NPD/ITO stacks clearly showed that the introduction of 2-nm-thick SOP interlayers induced a clear energy-level shift between the organic layers (Supplementary Fig. 15a, b). These results clearly indicate that the energy-level shift by the negative SOP interlayer induced (ca. 0.45 eV) reduces the injection barrier (HOMO level gap) between the NPD and the mCBP layers to improve the hole injection. Note that the author confirmed that the vacuum level shift at the NPD/mCBP interface was almost negligible (Supplementary Fig. 15c), and the surface potentials on the SOP interlayers were stable under vacuum conditions (Supplementary Fig. 15d). Additionally, the DCM results (Supplementary Fig. 16a) of these OLEDs indicated that the thin SOP interlayers induced no distinct charge accumulation in the OLEDs observed in the OLEDs with SOP-ETLs (Supplementary Fig. 14). However, the CEs with the SOP interlayers (Fig. 6d) decreased because of the change in the carrier balance, that is, the hole-rich situation. To improve the carrier balance of the OLED with the 6F-2PI interlayer, an electron-transport molecule, 4,6-bis(3,5-di(pyridin-4-yl)phenyl)-2-methylpyrimidine (B4PyMPM, Supplementary Fig. 16b)[42], with a high electron mobility ($1.0 \times 10^{-4}$ cm$^2$ V$^{-1}$ s$^{-1}$) was applied instead of the SF3-TRZ ETL. The CE of the OLED with the 6F-2PI interlayer was successfully improved using a B4PyMPM ETL, and the maximum CE (101.7 cd A$^{-1}$) was higher than that of the reference device (Fig. 6d and Supplementary Table 5) because of the well-tuned carrier balance. Therefore, the introduction of thin SOP layers between the organic/organic layers is beneficial for precisely tuning the carrier balance to simultaneously realize a low driving voltage and a high EL quantum efficiency.

In conclusion, the author developed polar molecules exhibiting a high GSP slope in vacuum-deposited films. The 6FDI backbone provided highly ordered PDM orientations and strong molecular polarization originating from the 6F and the PI units. The introduction of PI resulted in a strong molecular PDM and an increase in $T_g$, leading to improved film polarization. The presence of conformers directly influenced the effective magnitude of the molecular dipoles to form the film polarization of 6FDI-based molecules. Furthermore, the SOP polarity can be designed by controlling the dipole direction using the positional isomers of polar molecules. The resulting GSP slope of 6FDI-2$p$BN on an ITO substrate without $T_s$ control was approximately 280 mV nm$^{-1}$, which is 5.6 times higher than that of Alq$_3$. Furthermore, the OLED study revealed that thin SOP interlayers between organic/organic interfaces tune the carrier injection barrier, leading to improved device performance. The findings of this study provide design strategies for extremely strong SOP for applications in organic electronic and energy-harvesting devices.

## Methods

### Materials and general methods
All synthesis reagents were purchased from commercial sources (TCI) and used without further purification. All the synthesized compounds were purified by column chromatography followed by temperature-gradient vacuum sublimation. HAT-CN (>99.7% purity) was purchased from Analysis Atelier Corporation. NPD (>99.0% purity) and mCBP (>99.0% purity) were purchased from TCI. 4CzIPN (>98.0% purity), SF3-TRZ (>99.0% purity), and B4PyMPM (>99.0% purity) were purchased from Lumtec. Nuclear magnetic resonance (NMR) spectra were obtained using a JNM-ECX400 NMR spectrometer (JEOL) at ambient temperature. Absorption spectra of organic films on a quartz glass

substrate were measured on an UV-2550 (Shimadzu). A glass transition temperature was determined using differential scanning calorimetry on a DSC7000X (Hitachi).

### Film sample fabrication and evaluation
Organic solid-state films of varying thicknesses for surface potential measurements were deposited directly on pre-cleaned, 100-nm-thick, ITO-coated glass substrates using the physical vapor deposition (PVD) technique. PVD was performed under high vacuum at pressure levels $<3 \times 10^{-4}$ Pa at a monitored deposition rate using an in-house evaporation machine. The surface potential was measured using the Kelvin probe method under vacuum and dark conditions (UHVKP020, K.P. Technology). Film thickness was estimated using a thickness meter (FR-ES, ThetaMetrisis). The molecular density was assumed to be the same as that of 6F-2BN to estimate the average orientation degree of the permanent dipole moment in the film. To measure vibration-based generated current, a probe (stainless steel) with a diameter of 4 mm of a KP measurement system was placed above a deposited organic film on an ITO substrate with a gap of -1 mm, and the probe was vibrated with a frequency of 59.2 Hz, then the generated current was collected with an oscilloscope (TBS2104B, Tektronix) using a current/voltage amplifier (SA-604F2, NF). The HOMO levels were estimated using a photoelectron yield spectrometer (PYS). The PYS measurements of 100-nm-thick films on ITO-coated substrates were performed using AC-3 (RIKEN KEIKI) and BIP-KV100 (Bunkou-keiki). Because of the deep HOMO levels of the molecules, the experimental results for BIP-KV100 were used to estimate the HOMO levels.

### Device fabrication and characterization
OLEDs, HODs, and EODs were fabricated by the vacuum vapour deposition process without exposure to ambient air. After fabrication, devices were immediately encapsulated under a glass cover using epoxy glue in a nitrogen-filled glovebox (H$_2$O > 0.1 ppm, O$_2$ > 0.1 ppm). All organic layers were deposited at a deposition rate of 0.1 nm s$^{-1}$. The deposition rate of LiF and Al layers was 0.01 and 0.5 nm s$^{-1}$. All device characterizations were performed at room temperature. Current density–luminance–voltage measurements of OLEDs were performed using a sourcemeter (Keithley 2400) and a luminance meter (LS160, KONICA MINOLTA). Current density-luminance characteristics of HODs and EODs were measured using a sourcemeter (Keithley 2400) For DCM, repeated-triangular voltage signals were applied to each device using a function generator (AFG1062, Tektronix) and the displacement current amplified using a current amplifier (CA5351, NF). The applied voltage and amplified current were measured using an oscilloscope (WaveSurfer 4054HD, Teledyne Lecroy). The applied voltage scan rate was 2k V s$^{-1}$. PL and EL spectra were collected using a spectrameter (Flame-T, Ocean Photonics). Transient PL and EL profiles were collected using a photomultiplier tube (H10721-01, Hamamatsu Photonics), a current/voltage amplifier (SA-604F2), and an oscilloscope (WaveSurfer 4054HD).

### Computational calculations
Optimized molecular structures and permanent dipoles of ground-state molecules were calculated using the B3LYP/6-31 G (d) level with the Gaussian 16 program package. Conformation analysis was performed using force-field theory with CONFLEX 9.

### Synthesis
6FDI-2BTA: 4,4′-(hexafluoroisopropylidene)diphthalic anhydride (621 mg), 6-aminobenzothiazole (430 mg), and molecular sieve (4A) were added to dehydrated 1,3-dimethyl-2-imidazolidinone (5 mL). The solution was stirred at room temperature for 20 min and subsequently stirred at 150 °C for 20 h. The resulting solution was washed with water. The precipitate was dried under vacuum, and purified by

chromatography on silica gel (chloroform-ethyl acetate-hexane) to afford 6FDI-2BTA as a white solid in 60% yield. $^1$H NMR (400 MHz, CDCl$_3$): $\delta$ 9.09 (s, 2H), 8.28 (d, $J = 8.7$ Hz, 2H), 8.10 (d, $J = 8.2$ Hz, 2H), 8.07 (d, $J = 1.9$ Hz, 2H), 7.97 (s, 2H), 7.94 (d, $J = 8.2$ Hz, 2H), 7.49 (dd, $J = 2.0, 8.8$ Hz, 2H). $^{13}$C NMR (100 MHz, DMSO-d$^6$): $\delta$ 166.76, 166.63, 158.48, 153.17, 140.93, 137.94, 136.46, 134.38, 133.60, 133.19, 129.50, 126.49, 125.04, 124.23, 123.73, 122.29. HRMS (ESI) m/z: [M + H]$^+$ calcd. for C$_{33}$H$_{15}$F$_6$N$_4$O$_4$S$_2$ 709.0433; found 709.0440.

6FDI-2TAZ: 4,4′-(hexafluoroisopropylidene)diphthalic anhydride (621 mg), 1-(4-aminophenyl)-1,2,4-triazole (450 mg), and molecular sieve (4A) were added to 1,3-dimethyl-2-imidazolidinone (5 mL). The solution was stirred at room temperature for 20 min and subsequently stirred at 150 °C for 20 h. The resulting solution was washed with water. The precipitate was dried under vacuum, and purified by chromatography on silica gel (chloroform-ethyl acetate-hexane) to afford 6FDI-2TAZ as a white solid in 56% yield. $^1$H NMR (400 MHz, CDCl$_3$): $\delta$ 8.61 (s, 2H), 8.14 (s, 2H), 8.09 (d, $J = 7.8$ Hz, 2H), 7.93 (d, $J = 8.7$ Hz, 4H), 7.86 (d, $J = 8.7$ Hz, 4H), 7.64 (d, $J = 9.1$ Hz, 4H). $^{13}$C NMR (100 MHz, DMSO-d$^6$): $\delta$ 166.55, 166.41, 161.54, 144.57, 144.57, 143.19, 137.92, 136.82, 133.58, 133.17, 131.44, 130.95, 129.29, 125.04, 124.25, 122.40, 120.52. HRMS (ESI) m/z: [M + H]$^+$ calcd. for C$_{35}$H$_{19}$F$_6$N$_8$O$_4$ 729.1428; found 729.1434.

6FDI-2pBN: 4,4′-(hexafluoroisopropylidene)diphthalic anhydride (621 mg), 4-aminobenzonitrile (332 mg), and molecular sieve (4 A) were added to 1,3-dimethyl-2-imidazolidinone (5 mL). The solution was stirred at room temperature for 20 min and subsequently stirred at 150 °C for 20 h. The resulting solution was washed with water. The precipitate was dried under vacuum, and purified by chromatography on silica gel (chloroform-ethyl acetate-hexane) to afford 6FDI-2pBN as a white solid in 80% yield. $^1$H NMR (400 MHz, CDCl$_3$): $\delta$ 8.09 (d, $J = 8.2$ Hz, 2H), 7.95 (s, 2H), 7.93 (d, $J = 8.2$ Hz, 2H), 7.83 (d, $J = 8.7$ Hz, 4H), 7.66 (d, $J = 8.7$ Hz, 4H). $^{13}$C NMR (100 MHz, CDCl$_3$): $\delta$ 165.39, 165.20, 139.65, 136.42, 135.43, 133.21, 132.42, 132.12, 126.61, 125.74, 124.66, 118.13, 112.02. HRMS (ESI) m/z: [M+Na]$^+$ calcd. for C$_{33}$H$_{14}$F$_6$N$_4$NaO$_4$ 667.0811; found 667.0823.

6FDI-2pBNMe: 4,4′-(hexafluoroisopropylidene)diphthalic anhydride (622 mg), 4-amino-3-methylbenzonitrile (370 mg), and molecular sieve (4A) were added to 1,3-dimethyl-2-imidazolidinone (5 mL). The solution was stirred at room temperature for 20 min and subsequently stirred at 150 °C for 20 h. The resulting solution was washed with water. The precipitate was dried under vacuum, and purified by chromatography on silica gel (chloroform-ethyl acetate) to afford 6FDI-2pBNMe as a white solid in 72% yield. $^1$H NMR (400 MHz, CDCl$_3$): $\delta$ 8.09 (d, $J = 8.3$ Hz, 2H), 7.96 (d, $J = 8.2$ Hz, 2H), 7.93 (s, 2H), 7.70 (s, 2H), 7.66 (dd, $J = 1.4, 8.2$ Hz, 2H), 7.34 (d, $J = 8.2$ Hz, 2H), 2.29 (s, 6H). $^{13}$C NMR (100 MHz, CDCl$_3$): $\delta$ 165.50, 165.24, 139.53, 138.40, 136.28, 135.07, 134.53, 132.75, 132.45, 130.78, 129.75, 125.76, 124.65, 118.05, 113.76, 18.34. HRMS (ESI) m/z: [M+Na]$^+$ calcd. for C$_{35}$H$_{18}$F$_6$N$_4$NaO$_4$ 695.1124; found 695.1116.

6FDI-2oBN: 4,4′-(hexafluoroisopropylidene)diphthalic anhydride (746 mg), 2-aminobenzonitrile (397 mg), benzoic acid (205 mg), and molecular sieve (4 A) were added to 1,3-dimethyl-2-imidazolidinone (6 mL). The solution was stirred at room temperature for 20 min and subsequently stirred at 150 °C for 20 h. The resulting solution was washed with water. The precipitate was dried under vacuum, and purified by chromatography on silica gel (chloroform-ethyl acetate) to afford 6FDI-2oBN as a white solid in 75% yield. $^1$H NMR (400 MHz, CDCl$_3$): $\delta$ 8.10 (d, $J = 7.8$ Hz, 2H), 8.06 (s, 2H), 7.88 (d, $J = 8.2$ Hz, 4H), 7.79 (t, $J = 7.8$ Hz, 2H), 7.62 (t, $J = 7.6$ Hz, 2H) 7.49 (d, $J = 7.8$ Hz, 2H). $^{13}$C NMR (100 MHz, CDCl$_3$): $\delta$ 165.48, 165.29, 139.61, 134.40, 132.42, 132.36, 132.12, 131.81, 130.57, 130.31, 129.69, 125.74, 124.65, 117.84, 113.65. HRMS (ESI) m/z: [M+Na]$^+$ calcd. for C$_{33}$H$_{14}$F$_6$N$_4$NaO$_4$ 667.0811; found 667.0823.

6FDI-2mBN: 4,4′-(hexafluoroisopropylidene)diphthalic anhydride (746 mg), 3-aminobenzonitrile (397 mg), benzoic acid (205 mg), and molecular sieve (4A) were added to 1,3-dimethyl-2-imidazolidinone (6 mL). The solution was stirred at room temperature for 20 min and subsequently stirred at 150 °C for 20 h. The resulting solution was washed with water. The precipitate was dried under vacuum, and purified by chromatography on silica gel (chloroform-ethyl acetate) to afford 6FDI-2mBN as a white solid in 72% yield. $^1$H NMR (400 MHz, CDCl$_3$): $\delta$ 8.10 (d, $J = 7.8$ Hz, 2H), 7.95 (s, 2H), 7.93 (d, $J = 8.3$ Hz, 2H), 7.82 (s, 2H), 7.74 (m, 4H), 7.66 (dd, $J = 7.8, 8.2$ Hz, 2H). $^{13}$C NMR (100 MHz, CDCl$_3$): $\delta$ 165.48, 165.29, 153.52, 139.61, 136.40, 132.42, 132.36, 132.12, 131.81, 130.57, 130.31, 129.69, 125.74, 124.65, 117.84, 113.65. HRMS (ESI) m/z: [M+Na]$^+$ calcd. for C$_{33}$H$_{14}$F$_6$N$_4$NaO$_4$ 667.0811; found 667.0813.

6F-2PI: Phthalic anhydride (550 mg), 4,4′-(hexafluoroisopropylidene)dianiline (561 mg), benzoic acid (410 mg), and molecular sieve (4A) were added to 1,3-dimethyl-2-imidazolidinone (6 mL). The solution was stirred at room temperature for 20 min and subsequently stirred at 150 °C for 20 h. The resulting solution was washed with water. The precipitate was dried under vacuum, and purified by chromatography on silica gel (chloroform-ethyl acetate) to afford 6F-2PI as a white solid in 75% yield. $^1$H NMR (400 MHz, CDCl$_3$): $\delta$ 7.99 (dd, $J = 3.2, 5.5$ Hz, 4H), 7.82 (dd, $J = 3.2, 5.5$ Hz, 4H), 7.57 (m, 8H). $^{13}$C NMR (100 MHz, CDCl$_3$): $\delta$ 167.01, 134.74, 132.73, 132.57, 131.65, 131.11, 125.93, 124.03, 122.69, 64.41. HRMS (ESI) m/z: [M+Na]$^+$ calcd. for C$_{31}$H$_{16}$F$_6$N$_2$NaO$_4$ 617.0906; found 617.0910.

## Data availability

The experimental data in this study are provided in the Source Data file. Additional data are available from the corresponding author upon request. Source data are provided with this paper.

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

## Acknowledgements

The author thanks Prof. Chihaya Adachi, Prof. Hajime Nakanotani, and Prof. Kenichi Goushi of Kyushu University for the setup of the OLED fabrication machine. The author appreciates Prof. Nobuhumi Nakamura and Prof. Takahiro Ichikawa of Tokyo University of Agriculture and Technology for their help and discussions. The author also thanks Prof. Keiichi Noguchi of Tokyo University if Agriculture and Technology for high resolution mass spectroscopy measurements, and Bunkoukeiki Co., Ltd. and RIKEN KEIKI Co., Ltd. for photoelectron yield spectroscopy measurements. This work was partially supported by JST FOREST Program (JPMJFR223S (M.T.)), JSPS KAKENHI (JP23H05406 and JP23K13716 (M.T.)), Inamori Foundation (M.T.), Tokuyama Science Foundation (M.T.), Casio Science Promotion Foundation (M.T.), and Advanced Technology Institute Research Grants (M.T.).

## Author contributions

The project was conceived and designed by M.T. M.T. synthesized the molecules and evaluated their properties.

## Competing interests

The authors declare no competing interests.
