## [Peer Review File · Nature Communications]

REVIEWER COMMENTS

Reviewer #1 (Remarks to the Author):

This is a nice study that builds on from the author's Nature Materials paper published in 2022. The novelty of the work presented in the present manuscript lies mainly in a series of new derivatives with tuneable polarity (permanent dipole moment) which in turn is used to deposit organic films with a high degree of surface orientation polarisation.

The work is carried out thoroughly and offers some new insight into this field and contributes to the development of structure-property relations. It is somewhat incremental in terms of contribution to scientific knowledge. Without being an expert in this specific area, I am not well placed to judge whether the achieved figures of merit (potential slopes in particular) are highly significant and impactful or not.

One issue I would like to raise is the "organic semiconductor" term used in the title. I could not find any mention of the semiconducting properties of these molecules. If these oriented organic films are to be used in organic electronic devices, I assume that they will need to have some semiconducting properties e.g. if used as an electron- or hole-transporting layer. The author should thus investigate the semiconducting properties of these organic films.

Lastly, all new materials are only characterised by ^1H NMR which is insufficient characterisation for novel compounds. Please include ^{13}C NMR and HRMS or other characterisation technique in line with typical organic chemistry journal recommendations. Please also include the actual ^1H and ^{13}C NMR spectra in the supporting information, this is often immensely helpful for researchers replicating or continuing work in an area.

Reviewer #2 (Remarks to the Author):

The submitted manuscript is a follow-up on a previous publication (Ref. 22) by the same author (and others). The main improvement in this work now is an introduction of a new functional group into the already established molecular structure, which leads to enhanced dipole moments, and, simultaneously to higher glass transition temperatures of the newly synthesized compounds. Both effects boost the giant surface potential of evaporated films to a new record of almost 300 mV/nm --- values that could be achieved before only by substrate cooling. As such, the work is original and of high quality and is suitable for publication if the following issues have been clarified:

- 1) Both in the abstract and the conclusion, the author claims application of the materials in (opto)electronic devices. However, there is no such device studies in this work. It is not even clear if these materials are suitable for electronic devices, regarding their large band gaps. There is no data on their energy levels (HOMO, LUMO) and their charge transport properties. The only kind of device application that is demonstrated, is a vibration energy harvesting device. But this has been shown before by other groups as well.
- 2) Fig. 1d contains a bunch of data points taken from literature that are not labeled (and not referenced) --- basically all the black dots. This is useless, because apart from the author nobody would be able to assign these data to a known material. Thus, all data need to be labeled (or summarized in a separate

table so that they can be identified).

3) Both Table 1 of the manuscript and the SI present a surface charge density σ , but is explained it is obtained from the measured GSP, specifically which epsilon is used (and whether the epsilon was actually measured).

4) It is written in the SI that the synthesized compounds were characterized by NMR, but no spectra are shown. Along the same line, DSC data have to shown as well to verify the given glass transition temperatures.

5) Normalisation of the GSP in FIG. S2 of the SI to the room temperature data doesn't make sense to me because the two compounds have different T_g so that they are at different values of T_s/T_g . Wouldn't it be clearer to give the (non-normalized) absolute GSP values of both?

6) Supplementary Fig. S5 doesn't tell much because the measured current is a more or less arbitrary number. If neither device area, nor parameters like current amplification etc. are given, there is no real physical meaning of the displayed currents. Anyway, it is pretty obvious that a material with large GSP will be suitable for VEGs (as demonstrated before e.g. in Refs. 13&14).

7) Finally, all in all, the manuscript contains quite a number of typos, wrong figure caption numbers, and also some awkward expression. Thus, language polishing is recommended.

Reviewer #3 (Remarks to the Author):

The manuscript presents a thorough study on the development of polar organic molecules that exhibit spontaneous orientation polarization (SOP) and possess high surface potential. The authors propose a novel molecular design strategy incorporating hexafluoropropane (6F) units to facilitate molecular orientation and phthalimide units to induce strong molecular polarization. The combination of 6F units and phthalimide units is innovative and well-justified. This approach aims to achieve high surface potentials in vacuum-deposited films without substrate temperature control, which is a significant advancement in the field of organic optoelectronics and energy-harvesting devices.

However, I must note that it still presents several inconsistencies, and a careful revisions of data presentation and details should be performed.

1. The author claimed that the surface potential slope of SOP molecules is approximately 280 mV/nm without substrate temperature control. I am interested in knowing which high-conductivity commercial organic semiconductors have a surface potential that can be comparable to this value. 2. If organic field-effect transistors are prepared by SOP molecules, what value of mobility can be achieved?

3. The author claimed that charge mobilities and light-emission performances strongly depend on the molecular orientations. But high mobility and light-emission performances usually have different requirements for molecular orientation. What kind of orientation of SOP molecules are needed for both application requirements?

4. The author only obtained the surface potential but did not prepare the relevant materials into corresponding OLEDs or OFETs to further verify the effectiveness of the strategy. Please supplement this section to make the work more complete.

5. There are some mistakes needed to correct,

a) page 2 line 45 "OPV" is abbreviation, but without mentioning the full name.

b) In Table 1, "T" in T_g needs to be italicized.

Replay to the reviewers

Firstly, I sincerely thank all the reviewers for taking her/his precious time to evaluate my manuscript and providing insightful and helpful comments and suggestions. These are keys to helping the author to improve the quality of the manuscript. Based on their comments, I revised the manuscript and added some experimental results such as semiconductor properties of the developed polar molecules. The developed molecules could not enhance device performances as ETLs in OLEDs with a typical stack structure because of their deep HOMO and LUMO levels as an electron transport layer. However, the experimental results clearly indicated significant impact of SOP polarity in OLED performances. Furthermore, I examined the impact of thin dipolar layers between organic layers to control hole injection from a hole transport layer (HTL) to an emission layer. An introduced thin SOP interlayer with positive SOP clearly suppressed hole injection between organic layers, and the negative SOP interlayer clearly lowered driving voltage of the OLEDs. Finally, the combination of a negative SOP interlayer and high mobility ETL improved the driving voltage and the EL quantum efficiency, I believe that these results indicates that the SOP can provide a unique perspective on the performance of organic devices, which differs from the widely investigated molecular packings or orientations that lead to high charge mobilities and efficient light out-coupling.

Additionally, the author missed the chemical investigations such as ^{13}C NMR and HRMS in the original manuscript. Their results are added to the Supplementary Information (NMR: **Supplementary Fig. 17-30**).

The author added the additional sentences, the figures, and table as follows:

[Additional sentences]

- [Page 2, line 21] Thus, SOP can provide a unique perspective on the performance of organic devices, which differs from the widely investigated molecular packings or orientations that lead to high charge mobilities and efficient light out-coupling.
- [Page 11, line 5] The energy levels of the highest occupied molecular orbital (HOMO) were determined to be 7.1–7.9 eV (**Supplementary Table 3**) using a photoelectron yield spectrometer (**Supplementary Fig. 7**). The lowest unoccupied molecular orbital (LUMO) levels were estimated using the HOMO levels and the optical gap values (**Supplementary Table 3**). The developed molecules possess deep HOMO and LUMO levels compared to typical electron transport molecules, such as Alq₃, because of the acceptor-rich 6F-based molecular structures. The vacuum-deposited films exhibited low-intensity photoluminescence (PL) (**Supplementary Fig. 8(a)-(g)**). The shapes of the PL spectra were relatively broad,

and the PL peak wavelength was approximately 500 nm. These are mainly attributed to charge-transfer (CT)-type emissions because the HOMO and LUMO distributions are spatially separated in the molecules. (**Supplementary Fig. 8(h)**). The carrier transport properties of the developed positive and negative SOP molecules, such as 6FDI-2*p*BN and 6FDI-2*o*BN, were characterized using hole-only and electron-only devices (HODs and EODs; device structures are shown in **Supplementary Fig. 9(a) and (b)**). The zero-field hole and electron mobilities (μ_{0h} and μ_{0e}) were estimated using the Child's law, that is, $J = \frac{9}{8} \mu_0 \epsilon_r \epsilon_0 \frac{V^2}{d^3}$, where μ_0 and d are the zero-field carrier mobility and thickness of organic films, respectively²⁹. The calculated μ_{0e} and μ_{0h} were $1-2 \times 10^{-9}$ and $1-2 \times 10^{-10}$ cm² V⁻¹ s⁻¹, respectively. The author estimates that the low carrier mobilities are attributed to the broad distribution of the density of states (DOS) owing to dipolar disorder and/or molecular conformations³⁰. Previous studies have revealed that large PDMs in deposited films induce a broad DOS, lowering the carrier mobilities^{16,31}. Furthermore, as discussed above, the 6FDI-backbone possesses several molecular conformations. **Supplementary Figs. 10 and 11** show the computationally calculated HOMO and LUMO energy alignments of the different conformations. Their energy levels are slightly different, which leads to a broadening of the DOS and a decrease in the charge mobilities.

- **[Page 11, line 29]** The author examined OLED performance using the developed dipolar films as electron-transport layers (ETLs). The OLED device structure (**Supplementary Fig. 12(a)**) based on a thermally activated delayed fluorescence (TADF) emitter, 1,2,3,5-tetrakis(carbazol-9-yl)-4,6-dicyanobenzene (4CzIPN), was 1,4,5,8,9,11-hexaazatriphenylenehexacarbonitrile (HAT-CN; 10 nm) / N,N'-di-1-naphthyl-N,N'-diphenylbenzidine (NPD; 30 nm) / 3,3'-di(9H-carbazol-9-yl)-1,1'-biphenyl (mCBP; 5 nm) / 15 wt% 4CzIPN:mCBP (30 nm) / ETL (50 nm) / LiF (1 nm) / Al (100 nm). 2-(9,9'-Spirobi[fluoren]-3-yl)-4,6-diphenyl-1,3,5-triazine (SF3-TRZ), 6FDI-2*o*BN, and 6FDI-2*p*BN were used as ETLs with nonpolar, negative SOP, and positive SOP, respectively. **Supplementary Fig. 12(b) - (d)** show the *J-V*-luminance (*L*), current efficiency (CE) profiles, and electroluminescence (EL) spectra of the OLEDs. The driving voltages of the OLEDs based on the 6FDI molecules were higher than that of the SF3-TRZ-based OLED. This was attributed to the low electron mobility of the 6FDI molecules (**Supplementary Table 4**) compared to SF3-TRZ³². Note that the deep LUMO levels of the 6FDI molecules allow electrons to be directly injected into doped 4CzIPN in the EML (**Supplementary Fig. 12(e)**). Because the EL spectra of the OLEDs mainly

attributed to 4CzIPN emission and the delayed EL lifetimes (4.1 μs) approximately correspond to the delayed PL lifetime (3.5 μs) of 4CzIPN doped in mCBP, electrons are directly injected to 4CzIPN molecules and charge recombination occurs in the EML (**Supplementary Fig. 13**). Although the SF3-TRZ-based OLED exhibited a maximum CE of 83.6 cd A^{-1} (corresponding to an external EL quantum efficiency of $\sim 25\%$)^{32,33}, those of the OLEDs based on 6FDI-2*o*BN and 6FDI-2*p*BN were 30.2 and 1.9 cd A^{-1} . The reason for the low CE was attributed to exciton quenching at the EML/ETL interface. The low electron mobilities of the 6FDI-based ETL and the high electron injection barrier between the ETLs and the EMLs facilitate hole accumulation and highly concentrated charge recombination at the interface, resulting in severe exciton-exciton and exciton-charge annihilations^{34–36}. Furthermore, the interfacial charge at the EML/ETL interface also induce charge accumulation. **Supplementary Fig. 14(a)** shows the displacement current measurement (DCM) profiles (ramp rate: 1 kV s^{-1}) of the OLEDs, where the current was measured using a current amplifier under triangular voltage applications. These results indicate that the ETLs with positive/negative SOPs induce the charge injection and accumulation at the lower voltage than the EL turn-on (**Supplementary Fig. 14(b) and (c)**). The negative/positive polarization charges at the EML/ETL interface induce hole/electron accumulations to induce exciton-charge annihilations lowering the EL quantum efficiency^{11,18,37,38}. The author estimated that the reason for the more severe CE drop of the OLED based on the 6FDI-2*p*BN ETL is the positive charge accumulation at the interface. The accumulated holes induce highly dense cations of 4CzIPN and mCBP, which possess absorption overlap with the 4CzIPN emission³⁵, indicating that 4CzIPN excitons are efficiently quenched by the cations compared to anions.

- **[Page 12, line 32]** A previous study investigated the impact of SOP layers on hole injection at the ITO/hole-transport layer (HTL) interface and revealed that an introduced negative SOP layer improves the hole current of hole-only devices²². This could be attributed to the adjustment of the work function using a dipole layer that reduces the carrier injection barrier, which has been well studied using self-assembled monolayers (SAMs)³⁹. Although SAM modifications can form a highly ordered and strong dipolar monolayer on metal electrodes, they are generally formed by a solution process method and are not applicable to organic/organic interfaces. On the other hand, the SOP layers formed by vacuum deposition are applicable to organic/organic and organic/metal interfaces. This study examined the impact of SOP interlayers on hole injection at organic/organic interfaces. In the OLED

structure shown in **Fig. 6(a)**, the relatively large HOMO level gap (~ 0.7 eV) between the NPD and the mCBP layers lowers the hole injection at the interface (**Fig. 6(b)**). To investigate the impact of the SOP layer at organic/organic interfaces, 2-nm-thick SOP layers of the developed polar molecules were introduced as interlayers at the interface between the NPD and the mCBP layers (**Supplementary Fig. 15(a) and (b)**). The reason for the small thickness of the SOP interlayers is the low hole mobility and the deep HOMO levels, reducing hole transport in OLEDs. Therefore, the author expects that the SOP interlayers act as interfacial dipoles to tune the interfacial energy differences⁴⁰. **Fig. 6(c)** shows the J - V - L characteristics of OLEDs with SOP interlayers. The OLED with the 6FDI-2pBN (positive SOP) interlayer exhibited a high driving voltage, whereas the OLED with the 6F-2PI (negative SOP) interlayer exhibited a low driving voltage compared to the OLED without interlayers. Furthermore, the thickness dependence of the surface potential of the mCBP/SOP-interlayer/NPD/ITO stacks clearly showed that the introduction of 2-nm-thick SOP interlayers induced a clear energy-level shift between the organic layers (**Supplementary Fig. 15(a) and (b)**). These results clearly indicate that the energy-level shift induced by the negative SOP interlayer (ca. 0.45 eV) reduces the injection barrier (HOMO level gap) between the NPD and the mCBP layers to improve the hole injection. Note that the author confirmed that the vacuum level shift at the NPD/mCBP interface was almost negligible (**Supplementary Fig. 15(c)**), and the surface potentials on the SOP interlayers were stable under vacuum conditions (**Supplementary Fig. 15(d)**). Additionally, the DCM results (**Supplementary Fig. 16(a)**) of these OLEDs indicated that the thin SOP interlayers induced no distinct charge accumulation in the OLEDs observed in the OLEDs with SOP-ETLs (**Supplementary Fig. 14**). However, the CEs with the SOP interlayers (**Fig. 6(d)**) decreased because of the change in the carrier balance, that is, the hole-rich situation. To improve the carrier balance of the OLED with the 6F-2PI interlayer, an electron-transport molecule, 4,6-bis(3,5-di(pyridin-4-yl)phenyl)-2-methylpyrimidine (B4PyMPM, **Supplementary Fig. 16(b)**)⁴¹, with a high electron mobility (1.0×10^{-4} cm² V⁻¹ s⁻¹) was applied instead of the SF3-TRZ ETL. The CE of the OLED with the 6F-2PI interlayer was successfully improved using a B4PyMPM ETL, and the maximum CE (101.7 cd A⁻¹) was higher than that of the reference device (**Fig. 6(d)**, **Supplementary Table 5**) because of the well-tuned carrier balance. Therefore, the introduction of thin SOP layers between the organic/organic layers is beneficial for precisely tuning the carrier balance to simultaneously realize a low driving voltage and a high EL quantum efficiency.

[Additional Figures and Tables]

Supplementary Table 3. Summary of E_{HOMO} , E_{LUMO} , and E_{gap} .

	E_{HOMO} (eV) ^{a)}	E_{LUMO} (eV) ^{b)}	E_{gap} (eV) ^{c)}
6FDI-2BTA	7.11	3.29	3.82
6FDI-2TAZ	7.14	3.27	3.87
6FDI-2 p BN	7.76	3.92	3.84
6FDI-2 m BN	7.72	3.89	3.83
6FDI-2 o BN	7.87	3.98	3.89
6F-2PI	7.56	3.80	3.76
6FDI-2 p BNMe	7.72	3.81	3.91

a) Energy level of HOMO. b) energy level of LUMO. c) Estimated from optical bandgap energy.

Supplementary Fig. 7. Photoelectron yield spectroscopy results. (a) 6FDI-2BTA. (b) 6FDI-2TAZ. (c) 6FDI-2*p*BN. (d) 6FDI-2*p*BNMe. (e) 6FDI-2*m*BN. (f) 6FDI-2*o*BN. (g) 6F-2PI.

Supplementary Fig. 8. Photoluminescence spectra. **(a)** 6FDI-2BTA. **(b)** 6FDI-2TAZ. **(c)** 6FDI-2pBN. **(d)** 6FDI-2pBNMe. **(e)** 6FDI-2mBN. **(f)** 6FDI-2oBN. **(g)** 6F-2PI. Vacuum-deposited films were excited using 365 nm-LED. **(h)** Calculated HOMO and LUMO distributions of 6FDI-2pBN and 6F-2PI.

Supplementary Fig. 9. Carrier transport properties. **(a)** Hole-only device (HOD) structure. **(b)** Electron-only device (EOD) structure. **(c)** and **(d)** Current density-voltage (V) characteristics of HODs based on 6FDI-2*p*BN **(c)** and 6FDI-2*o*BN **(d)**. **(e)** and **(f)** Current density- V characteristics of EODs based on 6FDI-2*p*BN **(e)** and 6FDI-2*o*BN **(f)**. V_{bi} represents built-in potential.

Supplementary Table 4. Summary of zero-field hole and electron mobilities (μ_{0h} and μ_{0e}) estimated using the Child's law.

	μ_{0h} ($\text{cm}^2 \text{V}^{-1} \text{s}^{-1}$)	μ_{0e} ($\text{cm}^2 \text{V}^{-1} \text{s}^{-1}$)
6FDI-2 p BN	6.8×10^{-10}	1.1×10^{-9}
6FDI-2 o BN	7.6×10^{-10}	5.7×10^{-9}

Supplementary Fig. 10. Computational calculated HOMO and LUMO levels of 6FDI-2pBN. **(a)** HOMO and LUMO distributions. **(b)** Energy alignment of LUMO+1, LUMO, HOMO, and HOMO-1. The Δ values denote the energy level differences between the conformer #1 and each conformer.

Supplementary Fig. 11. Computational calculated HOMO and LUMO levels of 6FDI-2oBN. **(a)** HOMO and LUMO distributions. **(b)** Energy alignment of LUMO+1, LUMO, HOMO, and HOMO-1. The Δ values denote the energy level differences between the conformer #1 and each conformer. Note that 6FDI-2oBN possesses 35 conformers, thus, the three conformations were extracted to investigate the impact of molecular conformations on energy levels.

Supplementary Fig. 12. Device performance of OLEDs with SOP-ETLs. **(a)** Device structure and molecular structures used in the device. **(b)** J - V - L characteristics. **(c)** Current efficiency (CE) profiles. **(d)** EL spectra at the current density of 10 mA cm^{-2} . **(e)** HOMO-LUMO energy diagrams of OLEDs using ETLs with deep LUMO levels.

Supplementary Fig. 13. PL of 4CzIPN in SOP layers and mCBP:ETL exciplex formation. **(a)** PL spectra of 15 wt%-doped 4CzIPN in mCBP, 6FDI-2pBN, and 6FDI-2oBN. **(b)** PL spectra of mixed films of mCBP and 6FDI molecules (50 wt%). **(c)** Transient PL profiles of 15 wt%-doped 4CzIPN in mCBP, 6FDI-2pBN, and 6FDI-2oBN. **(d)** Transient PL profiles of mixed films of mCBP and 6FDI molecules (50 wt%). **(e)** Transient EL profiles of OLEDs with various ETL (applied voltage: 10 V). **(f)** Transient EL profiles of OLEDs with various ETL (applied voltage: 10 V) with an off-bias of -10 V to remove the effect of stored charges on the transient profiles.

Supplementary Fig. 14. DCM results. **(a)** Current density-voltage characteristics of OLEDs based on various ETLs measured using DCM. **(b)** and **(c)** Schematic of charge accumulation at the interfaces induced by polarization charges at the interface of ETLs with positive SOP (b) and negative SOP (c).

Fig. 6. OLEDs with SOP interlayers. (a) OLED structure. (b) HOMO-LUMO energy diagrams. (c) J - V - L characteristics. (d) Current efficiency profiles.

Supplementary Fig. 15. SOP interlayers. **(a)** Energy level diagram of mCBP/6FDI-2*p*BN/NPD and surface potential profiles of mCBP (15 nm) / 6FDI-2*p*BN (2 nm) / NPD (30 nm) stack on an ITO substrate. The SOP-induced vacuum level shift (Δ_{SOP}) was determined using surface potential measurements. For the energy diagram, vacuum level shifts except for the Δ_{SOP} were ignored for simplicity. **(b)** Energy level diagram of mCBP/6F-2PI/NPD and surface potential profiles of mCBP (15 nm) / 6F-2PI (2 nm) / NPD (30 nm) stack on an ITO substrate. **(c)** Surface potential profiles of mCBP (12 nm) / NPD (30 nm) stack on an ITO substrate. **(d)** Time dependence of surface potentials on NPD/ITO, 6FDI-2*p*BN/NPD/ITO, and 6F-2PI/NPD/ITO stacks.

Supplementary Fig. 16. DCM profiles of OLEDs with SOP-interlayers. **(a)** Current density-voltage characteristics of OLEDs based on various ETLs measured using DCM. **(b)** Molecular structure of B4PyMPM.

Supplementary Table 5. OLED performance.

	V_{on} (V) ^{a)}	V (V) ^{b)}	CE_{max} (cd A ⁻¹) ^{c)}	CE (cd A ⁻¹) ^{d)}
SOP ETL				
SF3-TRZ (reference)	4.5	10.8	83.6	35.9
6FDI-2 p BN	7.5	14.4	1.9	1.8
6FDI-2 o BN	4.5	13.5	30.4	25.5
SOP interlayer				
No interlayer (reference)	4.5	10.8	83.6	35.9

6FDI-2pBN	6.9	13.2	56.4	28.5
6F-2PI	4.2	10.2	35.2	20.7
6F-2PI (B4PyMPM)	3.9	9.9	101.8	39.8

a) Voltage at 1 cd m^{-2} (V_{on}). b) Voltage at 10 mA cm^{-2} . c) Maximum value of CE (CE_{max}).
d) CE value at 10 mA cm^{-2} .

Reviewer #1 (Remarks to the Author):

This is a nice study that builds on from the author's Nature Materials paper published in 2022. The novelty of the work presented in the present manuscript lies mainly in a series of new derivatives with tuneable polarity (permanent dipole moment) which in turn is used to deposit organic films with a high degree of surface orientation polarisation.

The work is carried out thoroughly and offers some new insight into this field and contributes to the development of structure-property relations. It is somewhat incremental in terms of contribution to scientific knowledge. Without being an expert in this specific area, I am not well placed to judge whether the achieved figures of merit (potential slopes in particular) are highly significant and impactful or not.

One issue I would like to raise is the "organic semiconductor" term used in the title. I could not find any mention of the semiconducting properties of these molecules. If these oriented organic films are to be used in organic electronic devices, I assume that they will need to have some semiconducting properties e.g. if used as an electron- or hole-transporting layer. The author should thus investigate the semiconducting properties of these organic films.

Lastly, all new materials are only characterised by ^1H NMR which is insufficient characterisation for novel compounds. Please include ^{13}C NMR and HRMS or other characterisation technique in line with typical organic chemistry journal recommendations. Please also include the actual ^1H and ^{13}C NMR spectra in the supporting information, this is often immensely helpful for researchers replicating or continuing work in an area.

[Reply]: The author thanks the reviewer's helpful comments. I added the semiconductor properties of the developed polar molecules, such as charge mobility (**Supplementary Fig. 9**), HOMO-LUMO levels (**Supplementary Table 4**), and OLED results (**Fig. 6 and Supplementary Fig. 12-16**). Unfortunately, their properties were not impressive, however, SOP interlayers between an organic/organic interface clearly improved charge injection and OLED performance (please see upper section, **Fig. 6**). Additionally, I added the measurement results of ^1H and ^{13}C NMR and HRMS results in the Supplementary Information (NMR: Supplementary Fig. 17-30).

Reviewer #2 (Remarks to the Author):

The submitted manuscript is a follow-up on a previous publication (Ref. 22) by the same author (and others). The main improvement in this work now is an introduction of a new functional group into the already established molecular structure, which leads to enhanced dipole moments, and, simultaneously to higher glass transition temperatures of the newly synthesized compounds. Both effects boost the giant surface potential of evaporated films to a new record of almost 300 mV/nm --- values that could be achieved before only by substrate cooling. As such, the work is original and of high quality and is suitable for publication if the following issues have been clarified:

[Comment 1]:

1) Both in the abstract and the conclusion, the author claims application of the materials in (opto)electronic devices. However, there is no such device studies in this work. It is not even clear if these materials are suitable for electronic devices, regarding their large band gaps. There is no data on their energy levels (HOMO, LUMO) and their charge transport properties. The only kind of device application that is demonstrated, is a vibration energy harvesting device. But this has been shown before by other groups as well.

[Reply]: The author thanks the reviewer's valuable comments. The original manuscript totally lacked the semiconductor properties. I added the semiconductor properties of the developed polar molecules, such as charge mobility (**Supplementary Fig. 9**), HOMO-LUMO levels (**Supplementary Table 4**), and OLED results (**Fig. 6 and Supplementary Fig. 12-16**). Unfortunately, their properties were not impressive, however, SOP interlayers between an organic/organic interface clearly improved charge injection and OLED performance (please see upper section, **Fig. 6**).

[Comment #2]:

2) Fig. 1d contains a bunch of data points taken from literature that are not labeled (and not referenced) --- basically all the black dots. This is useless, because apart from the author nobody would be able to assign these data to a known material. Thus, all data need to be labeled (or summarized in a separate table so that they can be identified).

[Reply]: I added the list of the permanent dipole moment and the GSP slope of reported polar molecules as **Supplementary Table 1**. The author added the a table as follows:

[Additional table]:

Supplementary Table 1. List of GSP slope values of the reported in literatures.

Polar molecule	PDM (Debye)	GSP slope (mV/nm)
a-NPD ¹	0.34	+5.3
TPBi ¹	2.0	+43
BAIq ¹	2.32	+25
OXD-7 ¹	3.77	+68
Alq ₃ ¹	4.4	+48
Ir(ppy) ₃ ¹	6.41	-3.6
mCP ¹	1.35	-3.9
BCP ¹	2.9	+33
Al(7-prq) ₃ ¹	3.76	-103
Al(q-Cl) ₃ ¹	3.81	+94
Gaq ₃ ¹	4.45	+47
Znq ₂ ¹	5.13	+5.8
4CzPN ¹	6.49	+40
DACT-II ¹	2.02	+13
Ir(ppy) ₂ (acac) ¹	2.53	+38
4CzIPN ¹	3.85	+51
Bpy-OXD ¹	3.86	+37
B3PyMPM ¹	4.29	+3
2CzPN ¹	7.04	+58
DCJTb ¹	15.5	+14.8
BCPO ²	3.5	+151
p -ethyl-TPBi ³	7.0	+141
6F-2TRZ ⁴	2.97	-108
6F-2Cz ⁴	0.98	-40
6F-TPA-TRZ ⁴	3.19	-46
6F-Cz-TRZ ⁴	2.53	-63
6F-2BN ⁴	3.16	+69
3F-3BN ⁴	3.02	+130
6F-Cz-TRZ ⁴	2.53	-61

[Comment #3]:

3) Both Table 1 of the manuscript and the SI present a surface charge density sigma, but is explained it is obtained from the measured GSP, specifically which epsilon is used (and whether the epsilon was actually measured).

[Reply]:

I added the equation to calculate surface charge densities using GSP slope values. In this manuscript, the relative permittivity of the deposited film was assumed to be 3.0 as typical organic semiconductor materials. The author added the additional sentences as follows:

[Additional sentence]:

- **[Page 5, line 22] Note that the surface charge density (σ) of the dipolar films (Table 1) were calculated using the following equation, $\sigma = (\text{GSP slope}) \times \epsilon_r \times \epsilon_0$, where ϵ_r and ϵ_0 are the relative permittivity, the dielectric constant of vacuum. The ϵ_r value was assumed to be 3.0 in all organic films²⁴.**

[Comment #4]:

4) It is written in the SI that the synthesized compounds were characterized by NMR, but no spectra are shown. Along the same line, DSC data have to shown as well to verify the given glass transition temperatures.

[Reply]:

I added the NMR and the DSC results in the Supplementary Information (NMR: **Supplementary Fig. 17-30**). The author added the figures as follows:

[Additional figures]:

Supplementary Fig. 2. Differential scanning calorimetry (DSC) measurement results. (a) 6FDI-2BTA. (b) 6FDI-2TAZ. (c) 6FDI-2*p*BN. (d) 6FDI-2*p*BNMe. (e) 6FDI-2*o*BN. (f) 6FDI-2*m*BN. (g) 6F-2PI. (h) 6FDA.

[Comment #5]:

5) Normalisation of the GSP in FIG. S2 of the SI to the room temperature data doesn't make sense to me because the two compounds have different T_g so that they are at different values of T_s/T_g . Wouldn't it be clearer to give the (non-normalized) absolute GSP values of both?

[Reply]: The author thanks the insightful comment. The figure was revised to show absolute GSP values. The author revised the figures as follows:

[Revised Figure]:

Supplementary Fig. 3. T_s dependence of GSP slope. **(a)** Substrate temperature (T_s) dependence of the GSP slopes of 6F-2BN and 6FDI-2pBN. **(b)** T_s /glass transition temperature (T_g) dependence of GSP slope.

[Comment #6]:

6) Supplementary Fig. S5 doesn't tell much because the measured current is a more or less arbitrary number. If neither device area, nor parameters like current amplification etc. are given, there is no real physical meaning of the displayed currents. Anyway, it is pretty obvious that a material with large GSP will be suitable for VEGs (as demonstrated before e.g. in Refs. 13&14).

[Reply]: I revised the figure to show the current density. To measure the performance, the author used the probe with a diameter of 4 mm (area: 12.57 mm²) placed above an organic film (area: 25×25 mm²). the signals were collected using an oscilloscope and a current/voltage amplifier. The author revised the sentences and the figures as follows:

[Revised sentence]:

- **[Supplementary Information]** To measure vibration-based generated current, a probe (stainless steel) with a diameter of 4 mm of a KP measurement system was placed above a deposited organic film on an ITO substrate with a gap of ~1 mm, and the probe was vibrated with a frequency of 59.2 Hz, then the generated current was collected with an oscilloscope (TBS2104B, Tektronix) using a current/voltage amplifier (SA-604F2 NF).
- **[Page11, line 1]** **Supplementary Fig. 6** shows the current profile generated via probe vibration above the deposited 6FDI-2pBN film (surface potential of ~20 V), indicating that the deposited film of the developed molecules is applicable for energy harvesters and vibration sensors as presented in previous studies^{13,14}.

[Revised Figure]

Supplementary Fig. 6. Vibration-based generated current profile of the vacuum-deposited electret film. The probe vibrated above the 6FDI-2pBN film (surface potential~20 V), and the generated current was collected using an oscilloscope and a current/voltage amplifier. The current values were normalized by the probe area (12.57 mm²) because the area of the deposited film (25×25 mm²) was larger than the probe area.

[Comment #7]:

7) Finally, all in all, the manuscript contains quite a number of typos, wrong figure caption numbers, and also some awkward expression. Thus, language polishing is recommended.

[Reply]: Thank you for pointing it out. I carefully checked and revised the manuscript to improve the quality.

Reviewer #3 (Remarks to the Author):

The manuscript presents a thorough study on the development of polar organic molecules that exhibit spontaneous orientation polarization (SOP) and possess high surface potential. The authors propose a novel molecular design strategy incorporating hexafluoropropane (6F) units to facilitate molecular orientation and phthalimide units to induce strong molecular polarization. The combination of 6F units and phthalimide units is innovative and well-justified. This approach aims to achieve high surface potentials in vacuum-deposited films without substrate temperature control, which is a significant advancement in the field of organic optoelectronics and energy-harvesting devices.

However, I must note that it still presents several inconsistencies, and a careful revisions of data presentation and details should be performed.

[Comment #1]:

1. The author claimed that the surface potential slope of SOP molecules is approximately 280 mV/nm without substrate temperature control. I am interested in knowing which high-conductivity commercial organic semiconductors have a surface potential that can be comparable to this value.

[Reply]: I added the list of the SOP molecules (Supplementary) shown in Fig. 1(d). The relationship between the SOP intensity and the conductivity of polar molecules are not well understood, however, molecular polarity disturbs the charge transport based on the dipolar disorder effect, which reduces the conductivity of molecules (explained in Comment #2). The author added the table as follows:

[Additional Table]

Supplementary Table 1. List of GSP slope values of the reported in literatures.

Polar molecule	PDM (Debye)	GSP slope (mV/nm)
a-NPD ¹	0.34	+5.3
TPBi ¹	2.0	+43
BAIq ¹	2.32	+25
OXD-7 ¹	3.77	+68
Alq ₃ ¹	4.4	+48
Ir(ppy) ₃ ¹	6.41	-3.6
mCP ¹	1.35	-3.9

BCP ¹	2.9	+33
Al(7-prq) ₃ ¹	3.76	-103
Al(q-Cl) ₃ ¹	3.81	+94
Gaq ₃ ¹	4.45	+47
Znq ₂ ¹	5.13	+5.8
4CzPN ¹	6.49	+40
DACT-II ¹	2.02	+13
Ir(ppy) ₂ (acac) ¹	2.53	+38
4CzIPN ¹	3.85	+51
Bpy-OXD ¹	3.86	+37
B3PyMPM ¹	4.29	+3
2CzPN ¹	7.04	+58
DCJTb ¹	15.5	+14.8
BCPO ²	3.5	+151
p -ethyl-TPBi ³	7.0	+141
6F-2TRZ ⁴	2.97	-108
6F-2Cz ⁴	0.98	-40
6F-TPA-TRZ ⁴	3.19	-46
6F-Cz-TRZ ⁴	2.53	-63
6F-2BN ⁴	3.16	+69
3F-3BN ⁴	3.02	+130
6F-Cz-TRZ ⁴	2.53	-61

[Comment #2]:

2. If organic field-effect transistors are prepared by SOP molecules, what value of mobility can be achieved?

[Reply]: I estimated the charge mobilities of 6FDI-2*p*BN and 6FDI-2*o*BN in HODs and EODs based on the Child's law (**Supplementary Table 4 and Fig. 9**). The hole and electron mobilities were approximately 10^{-10} and 10^{-9} cm² V⁻¹ s⁻¹, respectively. The reasons for the low charge mobilities were mainly broad distribution of density of state (DOS). The author estimates that the broad DOS would originate from both conformational and dipolar disorders. As shown in the manuscript, the developed molecules possess many conformers, which have slightly different HOMO and LUMO energy levels, leading to the broad DOS. Additionally, the dipolar effect of the polar

molecules also induces the broad DOS. Because this study examined the application of the dipolar films to OLEDs, the mobilities were measured in the vertical device structures of HODs and EODs. As the reviewer pointer out, further study would be necessary to understand the charge transport in the developed dipolar films because the charge transport direction of organic field transistors is differ from the tested devices in this study. The author added the additional sentences, the figures, and table as follows:

[Additional sentence]

- **[Page 11, line 15]** The carrier transport properties of the developed positive and negative SOP molecules, such as 6FDI-2*p*BN and 6FDI-2*o*BN, were characterized using hole-only and electron-only devices (HODs and EODs; device structures are shown in **Supplementary Fig. 9(a) and (b)**). The zero-field hole and electron mobilities (μ_{0h} and μ_{0e}) were estimated using the Child's law, that is, $J = \frac{9}{8} \mu_0 \epsilon_r \epsilon_0 \frac{V^2}{d^3}$, where μ_0 and d are the zero-field carrier mobility and thickness of organic films, respectively³⁰. The calculated μ_{0e} and μ_{0h} were $1-2 \times 10^{-9}$ and $1-2 \times 10^{-10} \text{ cm}^2 \text{ V}^{-1} \text{ s}^{-1}$, respectively. The author estimates that the low carrier mobilities are attributed to the broad distribution of the density of states (DOS) owing to dipolar disorder and/or molecular conformations³¹. Previous studies have revealed that large PDMs in deposited films induce a broad DOS, lowering the carrier mobilities^{16,32}. Furthermore, as discussed above, the 6FDI-backbone possesses several molecular conformations. **Supplementary Figs. 10 and 11** show the computationally calculated HOMO and LUMO energy alignments of the different conformations. Their energy levels are slightly different, which leads to a broadening of the DOS and a decrease in the charge mobilities.

[Additional Figures]

Supplementary Fig. 9. Carrier transport properties. **(a)** Hole-only device (HOD) structure. **(b)** Electron-only device (EOD) structure. **(c)** and **(d)** Current density-voltage (V) characteristics of HODs based on 6FDI-2*p*BN **(c)** and 6FDI-2*o*BN **(d)**. **(e)** and **(f)** Current density- V characteristics of EODs based on 6FDI-2*p*BN **(e)** and 6FDI-2*o*BN **(f)**. V_{bi} represents built-in potential.

[Additional Table]

Supplementary Table 4. Summary of zero-field hole and electron mobilities (μ_{0h} and μ_{0e}) estimated using the Child's law.

	μ_{0h} ($\text{cm}^2 \text{V}^{-1} \text{s}^{-1}$)	μ_{0e} ($\text{cm}^2 \text{V}^{-1} \text{s}^{-1}$)
6FDI-2 p BN	6.8×10^{-10}	1.1×10^{-9}
6FDI-2 o BN	7.6×10^{-10}	5.7×10^{-9}

[Comment #3]:

3. The author claimed that charge mobilities and light-emission performances strongly depend on the molecular orientations. But high mobility and light-emission performances

usually have different requirements for molecular orientation. What kind of orientation of SOP molecules are needed for both application requirements?

[Reply]: Sorry for the ambiguity. This study additionally examined the role of the dipolar films in OLEDs. The author claims that the dipolar films are applicable to interlayers between organic layers. The introduction of SOP interlayers tunes the energy difference between the organic layers, and modulates the charge injection. This study revealed that a negative SOP interlayer lowers the hole injection barriers (HOMO differences) and reduces the driving voltage of OLEDs. I estimate that the effect of the thin interlayers is mainly attributed to interfacial dipoles. Then, because of the small thickness of the introduced interlayer (2 nm in this study), the developed molecules worked well as the SOP interlayer, despite the deep HOMO levels and the low mobilities. Thus, I believe that these results indicate that the SOP can provide a unique perspective on the performance of organic devices, which is different from the widely studied molecular orientations leading to high charge mobilities and efficient light out-coupling. The author added the additional sentences and the figures as follows:

[Additional sentences]

- **[Page 2, line 21]** Thus, SOP can provide a unique perspective on the performance of organic devices, which differs from the widely investigated molecular packings or orientations that lead to high charge mobilities and efficient light out-coupling.
- **[Page 13, line 9]** To investigate the impact of the SOP layer at organic/organic interfaces, 2-nm-thick SOP layers of the developed polar molecules were introduced as interlayers at the interface between the NPD and the mCBP layers (**Supplementary Fig. 15(a) and (b)**). The reason for the small thickness of the SOP interlayers is the low hole mobility and the deep HOMO levels, reducing hole transport in OLEDs. Therefore, the author expects that the SOP interlayers act as interfacial dipoles to tune the interfacial energy differences⁴¹.

[Additional figures]

Supplementary Fig. 15. SOP interlayers. **(a)** Energy level diagram of mCBP/6FDI-2pBN/NPD and surface potential profiles of mCBP (15 nm) / 6FDI-2pBN (2 nm) / NPD (30 nm) stack on an ITO substrate. The SOP-induced vacuum level shift (Δ_{SOP}) was determined using surface potential measurements. For the energy diagram, vacuum level shifts except for the Δ_{SOP} were ignored for simplicity. **(b)** Energy level diagram of mCBP/6F-2PI/NPD and surface potential profiles of mCBP (15 nm) / 6F-2PI (2 nm) / NPD (30 nm) stack on an ITO substrate. **(c)** Surface potential profiles of mCBP (12 nm) / NPD (30 nm) stack on an ITO substrate. **(d)** Time dependence of surface potentials on NPD/ITO, 6FDI-2pBN/NPD/ITO, and 6F-2PI/NPD/ITO stacks.

[Comment #4]:

4. The author only obtained the surface potential but did not prepare the relevant materials into corresponding OLEDs or OFETs to further verify the effectiveness of the strategy. Please supplement this section to make the work more complete.

[Reply]: As shown in above sections, this study additionally examined the role of SOP in OLEDs. Unfortunately, the developed dipolar films did not enhance the OLED performance as ETLs owing to the deep HOMO-LUMO levels and the low electron

mobility. However, the author revealed that the SOP interlayer tune the hole injection between the organic interface as the interfacial dipoles. Although self-assembled monolayers also provide the tunability of charge injections between metal electrodes and organic layers, the SOP interlayer can be introduced at organic/organic interfaces. The author added the additional sentences and the figures as follows:

[Additional sentences]

- **[Page 12, line 32]** A previous study investigated the impact of SOP layers on hole injection at the ITO/hole-transport layer (HTL) interface and revealed that an introduced negative SOP layer improves the hole current of hole-only devices²². This could be attributed to the adjustment of the work function using a dipole layer that reduces the carrier injection barrier, which has been well studied using self-assembled monolayers (SAMs)⁴⁰. Although SAM modifications can form a highly ordered and strong dipolar monolayer on metal electrodes, they are generally formed by a solution process method and are not applicable to organic/organic interfaces. On the other hand, the SOP layers formed by vacuum deposition are applicable to organic/organic and organic/metal interfaces. This study examined the impact of SOP interlayers on hole injection at organic/organic interfaces. In the OLED structure shown in **Fig. 6(a)**, the relatively large HOMO level gap (~0.7 eV) between the NPD and the mCBP layers lowers the hole injection at the interface (**Fig. 6(b)**). To investigate the impact of the SOP layer at organic/organic interfaces, 2-nm-thick SOP layers of the developed polar molecules were introduced as interlayers at the interface between the NPD and the mCBP layers (**Supplementary Fig. 15(a) and (b)**). The reason for the small thickness of the SOP interlayers is the low hole mobility and the deep HOMO levels, reducing hole transport in OLEDs. Therefore, the author expects that the SOP interlayers act as interfacial dipoles to tune the interfacial energy differences⁴¹. **Fig. 6(c)** shows the *J-V-L* characteristics of OLEDs with SOP interlayers. The OLED with the 6FDI-2pBN (positive SOP) interlayer exhibited a high driving voltage, whereas the OLED with the 6F-2PI (negative SOP) interlayer exhibited a low driving voltage compared to the OLED without interlayers. Furthermore, the thickness dependence of the surface potential of the mCBP/SOP-interlayer/NPD/ITO stacks clearly showed that the introduction of 2-nm-thick SOP interlayers induced a clear energy-level shift between the organic layers (**Supplementary Fig. 15(a) and (b)**). These results clearly indicate that the energy-level shift induced by the negative SOP interlayer (ca. 0.45 eV) reduces the injection barrier (HOMO level gap) between the NPD and the mCBP layers to improve the

hole injection.

- [Page 14, line 18] Furthermore, the OLED study revealed that thin SOP interlayers between organic/organic interfaces tune the carrier injection barrier, leading to improved device performance.

[Additional Figure]

Fig. 6. OLEDs with SOP interlayers. (a) OLED structure. **(b)** HOMO-LUMO energy diagrams. **(c)** $J-V-L$ characteristics. **(d)** Current efficiency profiles.

[Comment #5]:

5. There are some mistakes needed to correct,

a) page 2 line 45 “OPV” is abbreviation, but without mentioning the full name.

b) In Table 1, “T” in Tg needs to be italicized.

[Reply]: Thank you for pointing it out. I carefully checked and revised the manuscript to improve the quality. The word, OPV, was removed.

REVIEWERS' COMMENTS

Reviewer #1 (Remarks to the Author):

The author has responded thoroughly to all reviewer comments and performed a number of additional experiments that are now included in this revised version of the manuscript. All reviewers aired some concern about the lack of semiconducting properties in the original manuscript making the title somewhat misleading. Additional work has now been carried out to investigate the semiconducting properties and unfortunately, the semiconducting properties of these materials are not particularly impressive. I feel that the poor performance metrics in this context significantly weakens the overall impact of this work and its suitability for a general journal like Nat Commun. The work in its current form seems more suitable for a more specialised journal.

Reviewer #2 (Remarks to the Author):

The author of this manuscript has performed a really comprehensive revision of the manuscript by adding a vast set of new data to the SI and also including their discussion in the main text. To my opinion, all of the questions raised by previous reviewers have been answered so that the manuscript can be accepted for publication now.

Reviewer #3 (Remarks to the Author):

The authors have addressed my comments adequately and I recommend publication in current form.

Reply to Reviewers

Reviewer #1 (Remarks to the Author):

The author has responded thoroughly to all reviewer comments and performed a number of additional experiments that are now included in this revised version of the manuscript. All reviewers aired some concern about the lack of semiconducting properties in the original manuscript making the title somewhat misleading. Additional work has now been carried out to investigate the semiconducting properties and unfortunately, the semiconducting properties of these materials are not particularly impressive. I feel that the poor performance metrics in this context significantly weakens the overall impact of this work and its suitability for a general journal like Nat Commun. The work in its current form seems more suitable for a more specialised journal.

[Reply]:

Thank you for your comments. I agree that the semiconductor properties of the developed polar molecules in the manuscript, such as charge mobilities and photoluminescence, are not impressive. However, I note that the main point of this study is the molecular design for the formation of spontaneous orientation polarization (SOP). In addition, introducing thin dipolar layers of the developed molecules clearly affected OLED performance, especially, a negative SOP layer improved the device performance due to the reduced charge injection barriers between organic/organic interfaces. Although the semiconductor properties of the molecules were moderate, they successfully improved device performance. As I wrote in the introduction section of the manuscript, the SOP can provide a unique perspective on device performance. I think that the features of the developed molecules, such as film polarization in as-deposited films, are unique and widely applicable to devices with vertical stacks of organic and metal layers like OLEDs. However, as you pointed out, the word “organic semiconductor molecules” in the original title may be misleading to the readers. I revised the title as follows:

[Original title]:

Boosting spontaneous orientation polarization of organic semiconductor molecules based on fluoroalkyl and phthalimide units

[Revised title]:

Boosting spontaneous orientation polarization of polar molecules based on fluoroalkyl and phthalimide units

Reviewer #2 (Remarks to the Author):

The author of this manuscript has performed a really comprehensive revision of the manuscript by adding a vast set of new data to the SI and also including their discussion in the main text.

To my opinion, all of the questions raised by previous reviewers have been answered so that the manuscript can be accepted for publication now.

Reviewer #3 (Remarks to the Author):

The authors have addressed my comments adequately and I recommend publication in current form.

[Reply]:

I sincerely thank the reviewers #2 and #3 for the positive assessment.